behaviour, ecology

*Episyrphus balteatus*, *Eupeodes corollae*, flight behaviour, insect migration, orientation, sun compass

**Authors for correspondence:**
Gao Hu
e-mail: hugao@njau.edu.cn
Jason W. Chapman
e-mail: j.chapman2@exeter.ac.uk

# Adaptive strategies of high-flying migratory hoverflies in response to wind currents

Boya Gao[1,2], Karl R. Wotton[2], Will L. S. Hawkes[2], Myles H. M. Menz[3,4,5,6], Don R. Reynolds[7,8], Bao-Ping Zhai[1], Gao Hu[1] and Jason W. Chapman[1,2]

[1]Department of Entomology, College of Plant Protection, Nanjing Agricultural University, Nanjing 210095, People's Republic of China
[2]Centre for Ecology and Conservation, University of Exeter, Penryn, Cornwall TR10 9FE, UK
[3]Department of Migration, Max Planck Institute of Animal Behaviour, Radolfzell, Germany
[4]Centre for the Advanced Study of Collective Behaviour, and [5]Department of Biology, University of Konstanz, Konstanz, Germany
[6]School of Biological Sciences, The University of Western Australia, Crawley, Western Australia, Australia
[7]Natural Resources Institute, University of Greenwich, Chatham, Kent ME4 4TB, UK
[8]Rothamsted Research, Harpenden, Hertfordshire AL5 2JQ, UK

KRW, 0000-0002-8672-9948; MHMM, 0000-0002-3347-5411; DRR, 0000-0001-8749-7491; GH, 0000-0002-1000-5687; JWC, 0000-0002-7475-4441

Large migrating insects, flying at high altitude, often exhibit complex behaviour. They frequently elect to fly on winds with directions quite different from the prevailing direction, and they show a degree of common orientation, both of which facilitate transport in seasonally beneficial directions. Much less is known about the migration behaviour of smaller (10–70 mg) insects. To address this issue, we used radar to examine the high-altitude flight of hoverflies (Diptera: Syrphidae), a group of day-active, medium-sized insects commonly migrating over the UK. We found that autumn migrants, which must move south, did indeed show migration timings and orientation responses that would take them in this direction, despite the unfavourability of the prevailing winds. Evidently, these hoverfly migrants must have a compass (probably a time-compensated solar mechanism), and a means of sensing the wind direction (which may be determined with sufficient accuracy at ground level, before take-off). By contrast, hoverflies arriving in the UK in spring showed weaker orientation tendencies, and did not correct for wind drift away from their seasonally adaptive direction (northwards). However, the spring migrants necessarily come from the south (on warm southerly winds), so we surmise that complex orientation behaviour may not be so crucial for the spring movements.

## 1. Introduction

Under fine weather conditions, the daytime atmosphere above land, up to heights of 1 km or more, is occupied by an enormous abundance, biomass and diversity of small airborne insects [1–5]. This mass aerial movement, invisible from the ground, was discovered 80 years ago by extensive aerial sampling campaigns [6,7]. The overwhelming perception of this process is of passive transport on the prevailing wind currents, leading to widespread usage of the term 'aerial plankton' to describe this diverse group of organisms, and a commonly held assumption that the movements represent 'accidental dispersal'. Such movements are considered by many researchers to be fundamentally different from the latitudinal 'to-and-fro' type of migration [8], which has been intensively studied in macro-insects (typically body masses greater than 100 mg), particularly day-flying butterflies and dragonflies, and night-flying noctuid moths [9].

**Figure 1.** (*a*–*e*) Circular histograms of (*a*) the migration track directions of hoverfly mass migrations (repeated from [5]), (*b*) downwind directions during all days, (*c*) downwind directions during non-mass migration events, (*d*) downwind directions during mass migrations, and (*e*) flight headings during mass migrations exhibiting a significant degree of common orientation. Each small filled circle indicates the mean direction during a mass migration or a non-mass migration event. The bearing of the arrow indicates the mean direction of the entire dataset, while its length is proportional to the clustering of the dataset around the mean. (Online version in colour.)

While it is true that the smallest members of the daytime aerial fauna (micro-insects with body masses less than 10 mg) have no control over their movement direction once airborne [3], the process is not entirely passive even for these tiny organisms—micro-insects (e.g. aphids) decide when to launch into the air and may also exert some influence over their altitude and when they land [4,10]— and so the epithet 'passive' is not entirely applicable. Moreover, a recent radar study of insect movement high above southern Britain discovered that huge numbers of insects which are only slightly larger than this smallest size class (body masses 10–70 mg, hereafter 'meso-insects') exhibit seasonally beneficial migration directions (approximately northwards in spring and southwards in autumn [3]), despite the fact that prevailing wind directions do not facilitate such a seasonal reversal in movement direction. This surprising result revealed that a significant proportion of the day-flying aerial fauna, previously considered by many to be randomly dispersing aerial plankton, is in fact engaged in 'true' (i.e. to-and-fro) latitudinal migration between successive breeding areas [8,9].

Owing to the enormous scale of these migrations, and the diversity of roles that the immigrants play, such seasonal movements are of great significance to ecosystem functioning [11]. In the skies above southern Britain alone, 8–16 billion meso-insects (200–450 tons of biomass) undergo to-and-fro migrations each year [3], with impacts on energy flows, pollination, pest control, crop damage and disease spread [5,11]. Similar but larger movements undoubtedly occur over warmer continental landmasses [12,13], and so this represents the largest synchronized movement of terrestrial animals on the globe, rivalling the greatest marine migrations in scale and ecological impact. However, even though this class of high-flying insect migrant is hugely abundant, their flight behaviours in relation to winds have hardly been studied in comparison to other, less abundant, groups of insects. For example, almost all behavioural studies of high-altitude insect migrants have focused on large nocturnal species, particularly noctuid moths and grasshoppers, in which radar and tethered-flight studies have revealed the compass mechanisms and orientation strategies they employ to achieve windborne transport in seasonally beneficial directions

[14–16]. Considering day-flying migrants, almost all studies have focused on large butterflies and dragonflies; these species often remain close to the ground where winds will have less influence on migration direction, and use a solar-based compass to maintain flight in adaptive directions [17,18]. By comparison, we know almost nothing of the flight behaviours and orientation mechanisms of the hugely abundant diurnal meso-insects, and there is an urgent need to improve our knowledge of these invisible but huge 'bioflows'.

One of the most abundant and important components of the day-flying meso-insects are hoverflies (Diptera: Syrphidae), particularly smaller species such as *Episyrphus balteatus* and *Eupeodes corollae*, hereafter collectively referred to as 'migrant hoverflies'. These species make annual migrations into and out of higher latitude regions in copious numbers [5,19,20] and often fly high above the ground during migration [3,5]. Migrant hoverflies are vitally important components of temperate ecosystems for a number of reasons: (i) they are extremely abundant constituents of foodwebs and are responsible for the long-range transport of vast amounts of energy, biomass and nutrients [5]; (ii) larvae are voracious predators of aphids and provide important crop pest control services [5]; and (iii) adults are important pollinators [5,21,22]. Crucially, their recent population trends are stable [5,23], seemingly bucking the general trend of steep declines among other beneficial insect groups [23,24]. A recent radar study above southern Britain demonstrated that 1–4 billion migrant hoverflies fly high above this region each year [5]. Migrant hoverflies displayed a clear seasonal reversal of their migration direction: this was directed towards north-northwest (mean track direction = 342°) in the spring and due south (180°) in the autumn [5]; to aid interpretation of the results of the current study, we reproduce this previous data here as figure 1*a*.

The self-powered airspeeds of migrant hoverflies (approx. 2–3 m s$^{-1}$ [19]) will typically be considerably slower than wind speeds at their migration altitudes of several hundred metres above ground, and thus, it seems unlikely that passive downwind transport could explain the seasonal patterns of direction observed in our previous study [5]. In the current study, we use data from 155 000 radar-detected migrant

hoverflies to test the following predictions: (i) migrant hover-flies have a seasonal 'preferred direction of movement' (PDM [25]), which changes between spring and autumn gener-ations; (ii) mass hoverfly migrations occur on days when tailwinds are reasonably close to their seasonal PDM; and (iii) migrant hoverflies exhibit beneficial orientation strategies which can correct for small amounts of drift and, in concert with effective tailwind selectivity, increase the probability of windborne transport in the preferred direction. These ana-lyses will allow us to identify the precise mechanisms by which hoverflies achieve migratory movements in seasonally beneficial directions.

## 2. Material and methods

### (a) Radar operating procedures

We studied the flight behaviour of migrant hoverflies greater than 150 m above ground level (a.g.l.) during their spring and autumn migrations of the 10 year period from 2000 until 2009 inclusive. The 'spring migration period' was defined as May and June, and the 'autumn migration period' as August and Sep-tember [5], and analyses were restricted to the daytime period, spanning 1 h after sunrise until 1 h before sunset, as in our pre-vious analyses [5]. Data were collected by vertical-looking entomological radars (VLRs) situated in southern Britain, in the following locations: Malvern, Worcestershire, in southwestern England from 2000 to 2003; Chilbolton, Hampshire, in south-central England from 2004 to 2009; and Rothamsted, Harpenden, Hertfordshire, in southeastern England from 2000 to 2009. The VLRs are described in detail elsewhere [14], but we provide a brief summary here. Individual insect targets flying directly overhead are interrogated, and the returned signals include information on the time of passage, body mass, flight altitude, aerial density, displacement speed and direction and flight head-ing for all individual insects of greater than 2 mg body mass that fly through the vertically pointing beam within the altitude range of 150–1200 m a.g.l.

### (b) Separation of hoverfly radar data

The criteria and justification for identifying migrant hoverflies among the plethora of VLR-detected insects in our database (greater than 10 million individual insect targets) is explained in detail elsewhere [5], but we provide a summary here. As described in [5], historical observations of migratory concentrations at coastal sites and through mountain passes [1], and aerial sampling obser-vations [2], indicate that in western Europe, the most abundant migrant hoverflies in our selected size-range (see below) are *Ep. balteatus* (the marmalade hoverfly) and *Eu. corollae*, two aphido-phagous species which are known to be migratory in northern Europe [5]. Radar signals produced by these two species of migrant hoverfly were separated from the other VLR signals using a combination of size and shape parameters contained in the radar signals, and which are characteristic of our study species, as used previously [5]. Firstly, we used body-mass data from speci-mens of *Ep. balteatus* caught in the UK [5] to characterize the mass range of this species, and this provided a value of $22.3 \pm 6.6$ mg (mean ± 1 s.d.). Consequently, we used a mass range of 15–28 mg as a first filter of the radar data, to select only targets which matched *Ep. balteatus* in body mass. As the body mass range of *Eu. corollae* overlaps with that of *Ep. balteatus*, but all other migratory hoverflies have different body masses [5], we can be con-fident that this pre-selection will comprise only our two target species. The second stage of our data filtering involved the use of radar reflectivity values [14] to select targets with the expected body shape criteria of our migratory hoverflies. The two principal

ventral-aspect radar cross-sections ($\sigma_{xx}$ and $\sigma_{yy}$) of wild-caught *Ep. balteatus* were previously measured by us [5] using an X-band linear polarization transmission rig [14]. Laboratory measurement of the ratio of $\sigma_{xx}/\sigma_{yy}$ produced a value of $8.8 \pm 4.0$ (mean ± 1 s.d.) [5], and, therefore, we filtered our dataset of hover-fly-sized targets to include only individuals with values of $\sigma_{xx}/\sigma_{yy}$ lying between 5 and 10, as these would all fall within the range of the laboratory measurements for our main study species. If migrant hoverflies fly with a body axis that has a substantial pitch angle relative to the horizontal, this would potentially effect the estimation of their $\sigma_{xx}/\sigma_{yy}$ ratio and thus lead to mis-classification of targets. We confirmed that hoverflies migrate with a horizontal body axis by examining frames from a video of a mass immigration of *Eristalis tenax* hoverflies arriving at the northeastern tip of Cyprus in spring 2019: mean difference between body pitch angle and a perfectly horizontal reference line was 0.3° ($n = 100$, circular s.d. = 1.42°, $r = 0.996$, see the elec-tronic supplementary material, figure S1). Our radar filtering procedure led to a total of 367 817 individual targets categorized as migrant hoverflies during 2628 separate 'migration occasions' (any location/date combination when at least one migrant hover-fly was detected). Our previous analyses [5] indicated that migratory movements during mid-summer (July) were randomly directed [5], and so for this study, we restricted our analyses to the spring (May and June) and autumn (August and September) migration periods when movements were highly directed [5]. Removing the mid-summer data reduced our hoverfly dataset to 209 298 individual migrant hoverflies detected on 2030 separate occasions (931 in spring, when hoverflies comprised 6.01% of all radar-detected day-flying insects; and 1099 in autumn, when hoverflies comprised 10.16% of all radar-detected day-flying insects; see the electronic supplementary material, table S1).

### (c) Analysis of insect direction and wind data

To analyse directional patterns, we used a subset of the data which we termed hoverfly 'mass migrations'. These were selected by: (i) ordering all 'migration occasions' (when one or more migrant hoverflies had been detected), in the spring and autumn separately, from the highest number of hoverflies detected to the lowest; (ii) categorizing all 'migration occasions' as 'mass migrations' until cumulatively 75% of the total number of individual hoverflies in that season was reached; and (iii) removing all remaining migration occasions from the directional analyses. This ensures that only mass migrations are used to identify seasonal migration patterns. This process resulted in 434 of the hoverfly migration occasions (21% of the 2030 total occasions) being defined as mass migrations accounting for 75% (155 538) of the individual hoverflies, with 186 in spring and 248 in autumn (electronic sup-plementary material, table S1). For every individual hoverfly that passed through the beam, the VLR automatically recorded the migratory track vector (i.e. the displacement speed and direction relative to the ground); both of these parameters are largely deter-mined by the wind vector but are also somewhat influenced by the insect's flight vector [26]. The VLR also routinely records the body alignment of each insect, which is a measure of the hoverfly's self-powered flight heading, but which contains a 180° ambiguity as the head-end cannot be distinguished from the tail-end from the radar signal alone. Previous studies have demonstrated that directed migrants, flying at altitude, nearly always orient relatively close to the downwind direction [14,27], and so, the true heading was selected from the two possible values by choosing the value which was closest to the downwind direction.

The mean migration track and the mean flight heading, plus associated circular statistics, were calculated for each of the 434 mass migrations of hoverflies, using the Rayleigh test of uniform-ity [28]. The following three parameters were calculated for each of the distributions of individual tracks and flight headings: (i) the mean direction; (ii) the mean vector length 'r' (a measure

Proc. R. Soc. B 287: 20200406

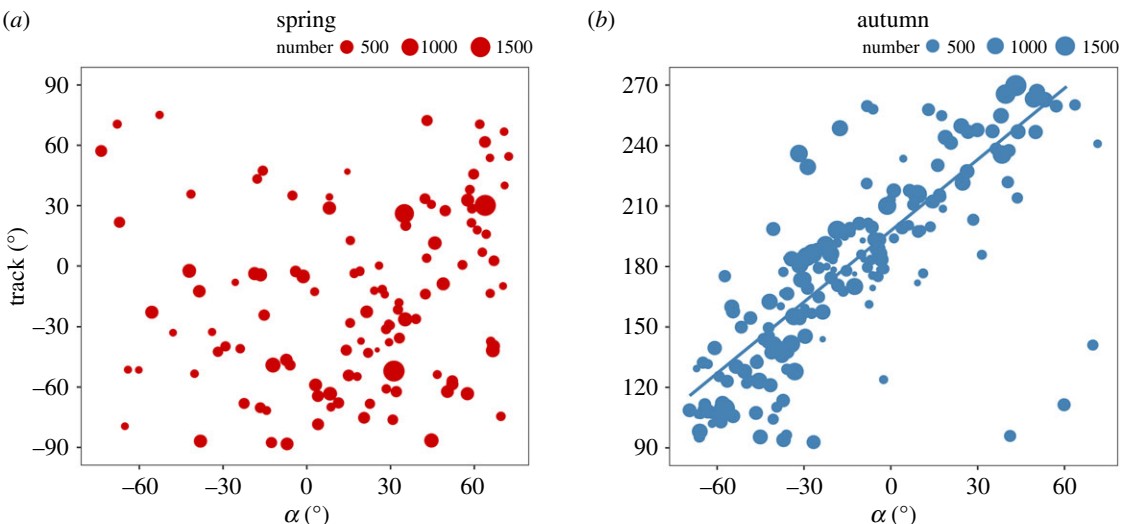

**Figure 2.** (*a*) Analyses of the extent of drift and degree of compensatory flight behaviour in spring (red dots) and (*b*) autumn (blue dots) hoverfly mass migrations. The mean track is plotted against $\alpha$ (the angle between track and heading) for each mass migration day, following Green & Alerstam [29], so that orientation responses to winds from different directions can be investigated. A weighted least-squares regression was calculated, and the line shows the change in track direction resulting from the combined effect of the downwind direction and the flight heading during the autumn (right panel). The number of individual hoverflies in each daily value is represented by the size of the dot.

of the clustering of the headings or tracks ranging from 0 to 1, with higher values indicating tighter clustering around the mean) for each distribution; and (iii) the probability that the distributions of tracks and headings differed from a uniform distribution (a *p*-value of less than 0.05 indicates that the distribution is significantly unimodal, and hence, the individuals in that mass migration event showed a significant degree of common alignment of their tracks and/or significant common orientation of their headings). All hoverfly migration events had significant common alignment of their track directions (reflecting the fact that they are strongly influenced by the wind which, during fair weather, is not expected to change its direction over the course of the day), but not all of them showed a significant degree of common orientation.

We then calculated the overall mean track and heading directions of the hoverfly mass migrations in the spring and autumn, by analysing the mean tracks and headings of the individual mass migrations with the Rayleigh test once again (figure 1). We did this first for each of the three radar locations separately, which showed that directional data were typically very similar across the three locations (electronic supplementary material, table S2 and figures S2 and S3). The majority of these datasets were not significantly different from each other when tested with the Watson–Wheeler test (electronic supplementary material, table S3), and when seasonal directions between locations were significantly different, the degree of difference was small (electronic supplementary material, table S2 and figures S2 and S3). We therefore pooled data from each of the radar locations for all further analyses. The seasonal distributions of track and heading directions when pooled across sites were significantly unimodal and roughly towards the north in spring and south in autumn. To investigate differences in orientation performance between spring and autumn, we carried out several comparisons of the heading *r*-values (from both the overall seasonal datasets) and from individual mass migration events.

To categorize the orientation response with respect to the flow, we calculated the angle $\alpha$ (the difference between the mean heading and the mean track [29], which we call the 'heading offset', see the electronic supplementary material, figure S4) for every hoverfly mass migration that exhibited a significant common orientation direction. The heading offset is a measure of the degree to which hoverflies attempted to correct for

wind-induced drift of their track away from their seasonal PDM [29]. The direction of $\alpha$ in each mass migration was categorized in two ways for further analyses. Firstly, $\alpha$ was categorized as either the heading being *anti-clockwise* of the track ($\alpha$ assigned a positive value) or *clockwise* of the track ($\alpha$ assigned a negative value, figure 2), and then the regression method of Green & Alerstam [29] was used to identify the orientation strategy employed with respect to the wind (figure 2). In this methodology, the value of the regression slope indicates the type of orientation strategy [26,29]: slope = 1 indicates a strategy of keeping a 'constant compass course' (also called 'full drift'); slope greater than 1 indicates an orientation strategy that we have previously termed 'compass-biased downstream orientation' (CBDO) [26], but which Green & Alerstam [29] refer to as 'overdrift'; slope less than 1 indicates 'partial compensation'; and slope = 0 indicates 'complete compensation'. The track direction at the intercept with $\alpha = 0$ is a measure of the seasonal PDM [29], which we used in the next stage of the analyses to determine the degree of correction for drift. We modified the methodology of Green & Alerstam by using the weighted least-squares linear regression method to calculate the slope and intercept, to account for the reasonably large differences in the number of individual hoverflies represented by each daily mean value (daily totals in mass migrations ranged from 38 to 1776 hoverflies). Furthermore, we restricted the circular data to a 180° semicircle (90°–270° in the autumn, and 90° to −90° in the spring), so that we could use linear regression methods. Secondly, $\alpha$ was categorized as either the heading being *closer* to the seasonal PDM than the track (in which case $\alpha$ was assigned a positive value) or *further away* from the seasonal PDM than the track (in which case $\alpha$ was assigned a negative value, see the electronic supplementary material, figure S4), and then an analysis was carried out to identify the degree of correction for lateral wind drift in each season (figure 3). We tested each distribution of $\alpha$ to determine if it was consistent with a mean angle of 0° (which would be expected if there was no systematic bias of $\alpha$ with respect to the downwind direction and the seasonal PDM), or if there was a significant positive (which would indicate significant correction for wind drift) or negative bias [26]. We used the Rayleigh test to calculate the mean value of $\alpha$ and the 95% confidence interval (CI) [28], and if they did not overlap with 0° and the mean value was positive, the situation was

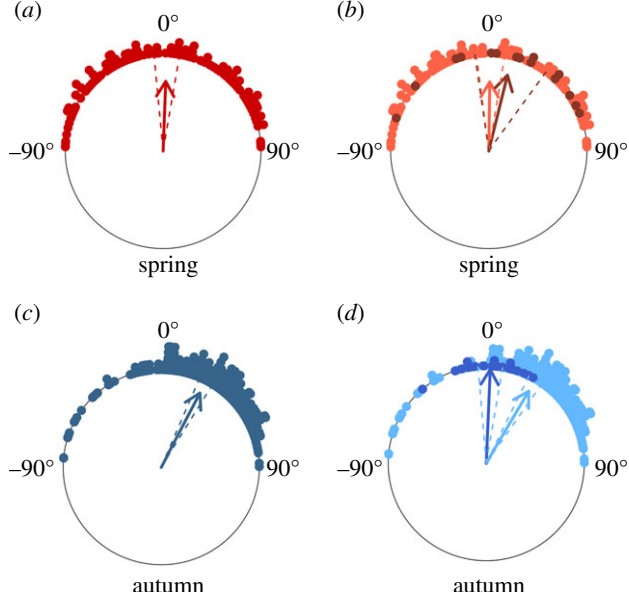

**Figure 3.** (*a–d*) Circular histograms of the distributions of the angle $\alpha$ (the difference between the mean heading and the mean track) during spring and autumn mass migrations of hoverflies. Small circles at the periphery of the plots represent values of $\alpha$ for each mass migration. An $\alpha$ of 0° indicates that the mean heading of the hoverflies was identical to the track on that event. Positive values of $\alpha$ (clockwise from 0°) indicated that hoverflies had a mean heading which was closer to the seasonal 'PDM' (spring: 0°; autumn: 198°) than the track, whereas negative values (anti-clockwise from 0°) indicated that headings were further away from the seasonal PDM than the track (see the electronic supplementary material, figure S4). The bearing of the arrow indicates the mean offset of the entire dataset, while its length is proportional to the clustering of the dataset around the mean; the dashed lines mark the 95% CI of the mean direction. (*a*) Values of $\alpha$ (red circles) for the entire spring dataset; (*b*) values of $\alpha$ for mass migrations when the track was less than 10° from the PDM (dark red circles) or greater than 10° from the PDM (light red circles) in the spring; (*c*) values of $\alpha$ (dark blue circles) for the entire autumn dataset; and (*d*) values of $\alpha$ for mass migrations when the track was less than 10° from the PDM (mid-blue circles) or greater than 10° from the PDM (light blue circles) in the autumn.

categorized as showing significant correction for wind drift [26,27]. This test was done separately for the total dataset in each season, for mass migrations when the angle $\Phi$ (the difference between the mean track and the seasonal PDM, which we call the 'drift angle' [29], see the electronic supplementary material, figure S4) was relatively small (less than 10°) in each season, and for when $\Phi$ was relatively large (greater than 10°, figure 3) in each season; previous studies on other species [27] have shown that an appreciable degree of drift (greater than 20°) is required before a compensatory response is induced.

Modelled estimates of downwind directions and speeds above our radar locations were acquired from the UK Meteorological Office website (https://www.metoffice.gov.uk/), for the spring and autumn periods of 2000–2009 at the same altitude (24 records d$^{-1}$) as our VLR sampling regime (150–1200 m a.g.l.). Seasonal downwind directions were also calculated using Rayleigh tests, and Watson–Wheeler tests were used to test for significant differences between the mean directions of insect and wind datasets [28]. Because wind speed data were not normally distributed, we used a non-parametric version of a two-way ANOVA (the Scheirer–Ray–Hare test [30]) to test for differences in wind speed between mass migration and non-mass migration days in spring and autumn, and any interaction between these factors.

## 3. Results

Our previous study of migrant hoverflies showed that they exhibit seasonally beneficial migration directions, towards 342° in the spring and due south in the autumn (results based on daily means from [5] and repeated here as figure 1*a*; directions were also very similar when calculated using all individual hoverflies rather than daily means, see the electronic supplementary material, figure S5). To test whether these seasonal directions could simply arise from a random selection of days for migration, we compared the distribution of downwind directions on all days: prevailing daytime winds averaged across the radar locations blew towards 61° in spring ($n = 907$, $r = 0.30$, $p < 0.001$) but in a more eastwards direction, towards 84°, in the autumn ($n = 977$, $r = 0.30$, $p < 0.001$; figure 1; electronic supplementary material, figures S2 and S3). Winds on days without hoverfly mass migrations (non-mass migrations) have similar mean directions to the total dataset in both seasons (spring: 73°, $n = 757$, $r = 0.32$, $p < 0.001$; autumn: 76°, $n = 773$, $r = 0.34$, $p < 0.001$; figure 1; electronic supplementary material, figures S2 and S3). If hoverflies selected a random subset of days for their mass migration events, then we should expect a similar pattern of wind directions to the non-mass migration days in both seasons. However, this was not the case: during the spring, hoverfly mass migrations occurred on downwind directions much closer to north (8°, $n = 150$, $r = 0.41$, $p < 0.001$; 65° closer to the PDM of 0° than the non-mass migration days; figure 1). Spring wind distributions on non-mass migration and mass migration days were significantly different from each other (Watson–Wheeler test: $W = 38.2$, $p < 0.001$), indicating active selection of favourably directed winds during spring mass migrations. In the autumn, the pattern was not so clear, as hoverfly mass migrations occurred on a considerably wider range of wind directions. However, there was a more southward component and a significant tendency towards the southeast on mass migration days (134°, $n = 204$, $r = 0.23$, $p < 0.001$; 58° closer to the PDM than the non-mass migration days; figure 1), which was significantly different from the winds on non-mass migration days ($W = 33.8$, $p < 0.001$). Thus, in the autumn, there is evidence for a certain degree of selectivity of winds (downwind directions towards the southeast versus east-northeast); however, the wind selectivity was not as pronounced as it was in the spring (compare the $r$-values of 0.23 versus 0.41), nor was the mean downwind direction as close to the seasonal PDM in the autumn as the corresponding values in spring. Given the apparently reduced level of wind selectivity in the autumn, and the general unsuitability of the winds in this season, how then do autumn migrants achieve migration in the seasonally beneficial direction?

The migrant hoverflies exhibited a significant degree of common orientation of their flight headings during a high proportion of mass migrations (electronic supplementary material, table S1). Similar to the migration tracks, the days with significant common orientation had mean headings in broadly seasonally beneficial directions: towards the northwest in the spring (314°, $n = 144$, $r = 0.57$ $p < 0.001$) and south-southwest in the autumn (198°, $n = 231$, $r = 0.78$, $p < 0.001$; figure 1). However, there were four important seasonal differences between the flight headings of spring and autumn migrants. Firstly, common orientation was notably more frequent in autumn (93% of occasions) than during spring

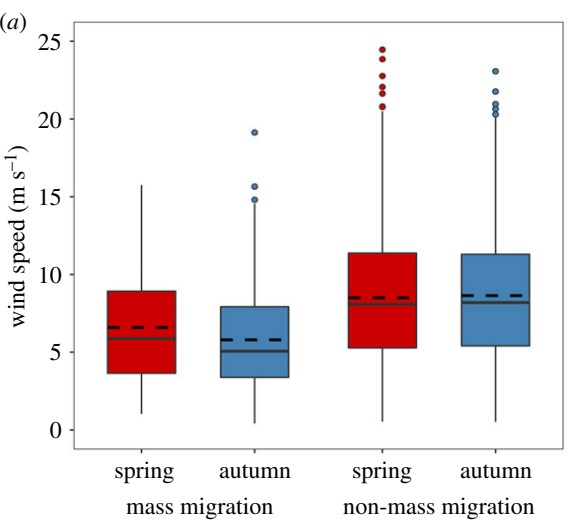

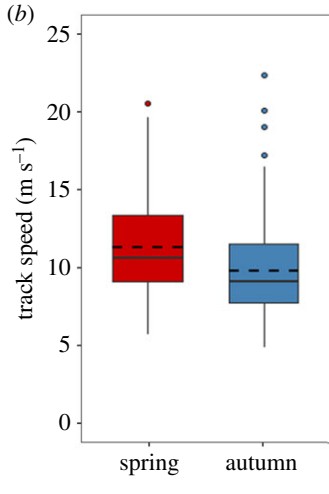

**Figure 4.** (*a*) The distribution of wind speeds during hoverfly mass migrations and non-migration days. (*b*) The distribution of displacement speeds during hoverfly mass migrations. The bottom and top of the boxes show the lower and upper quartile values, respectively. The horizontal solid black lines represent the median for each category, the dashed black lines represent the mean, whiskers indicate the 5th and 95th percentiles, while the small circles show the outliers. (Online version in colour.)

(73%). Secondly, the clustering of mean daily headings around the mean was notably tighter in autumn than spring (compare *r*-values of 0.78 versus 0.57 for the overall distribution), indicating a greater population-level preference (or capability) to orient in an advantageous direction in autumn. Thirdly, the clustering of individual headings around the mean during individual mass migrations was also significantly higher during the autumn (spring: mean *r*-value = 0.310; autumn: mean *r*-value = 0.498; *t*-test: $t_{430}$ = −13.62, $p < 0.001$; electronic supplementary material, table S1). Fourthly, the mean heading direction in autumn (198°) was identical to the seasonal PDM (see below), whereas the corresponding values in spring (314° and 0°, difference 46°) were rather different. Taken together, these results demonstrate that orientation performance was considerably higher in the autumn, with migrants seemingly more motivated to (and/or capable of) stronger orientation towards the seasonal PDM than during the spring.

Next, we investigated the patterns of the angle $\alpha$ (the 'heading offset') during spring and autumn in two related analyses. Firstly, using the method of Green & Alerstam [29], we plotted the mean track direction of each mass migration against $\alpha$ (in this case, $\alpha$ was assigned a positive or negative sign depending on whether the heading was anti-clockwise or clockwise of the track, respectively; see Material and methods). There was no clear relationship of track versus $\alpha$ in the spring, with a very wide scatter of points ($r^2$ = 0.03, $n$ = 114, $p$ = 0.07; figure 2*a*), indicating that spring hoverflies make no attempt to correct for drift. By contrast, the pattern of track against $\alpha$ in the autumn was very different, with a strong linear relationship and much less scatter ($r^2$ = 0.65, $n$ = 180, $p < 0.001$; figure 2*b*). The slope of 1.18 (95% CI = 1.05, 1.30) was significantly greater than 1 ($p < 0.05$), which indicates that the orientation strategy employed is the one we have previously termed CBDO [26]. The intercept of this relationship (the value of the track direction where $\alpha$ = 0, and thus the direction in which there is no correction) gave a value of 197.7° (95% CI = 193.0°, 202.4°) for the estimate of the seasonal PDM. We therefore used a value of 198° for the autumn PDM in the subsequent analyses. In order to look for evidence of time-compensation in the compass mechanism, we calculated the values of the mean track and mean heading of hoverfly mass migrations for each hour

of the day and compared the hourly values with the overall mean value across the whole day (see the electronic supplementary material, figure S6 and table S4). Time of day did not affect either directional measure (track: $r^2$ = 0.003, $n$ = 1190, $p$ = 0.088; heading: $r^2$ = 0.0004, $n$ = 1190, $p$ = 0.51); in both cases, hourly mean values remained very close to the overall daily mean value and there was no systematic pattern in the variation (electronic supplementary material, figure S6). This result indicates that hoverflies fully compensate for the Sun's apparent motion over the course of the day [31].

Secondly, we compared the frequency of mass migrations when the mean heading was closer to the seasonal PDM than the mean track (when $\alpha$ was assigned a positive value) with mass migrations where the heading was further away ($\alpha$ assigned a negative value). During the spring mass migrations, the mean value of $\alpha$ was not significantly different from 0°, and there were approximately equal numbers of positive and negative values, for both the total dataset, and the two subsets of data with small and large drift angles ($\Phi$), indicating that the direction of the heading was not related to the track and seasonal PDM (electronic supplementary material, table S5; figure 3*a*,*b*). By contrast, the autumn mass migrations showed a highly skewed distribution of $\alpha$, with 83% of the values in the total dataset being positive, and this distribution was significantly different from 0° with a mean value of +28° ($p < 0.05$, figure 3*c*). Further examination of the autumn data showed that values of $\alpha$ were random with respect to the track and PDM when drift angles were less than 10° (i.e. when the downwind was close to the PDM; figure 3*d*), but when $\Phi$ was greater than 10°, the hoverfly mass migrations showed a highly skewed distribution of positive values (87% of headings lay closer to the PDM than the tracks) with a mean value of +30° ($p < 0.05$, figure 3*d*). This result indicates that as the degree of wind-induced lateral drift away from the seasonal PDM in autumn (198°) increased, hoverflies showed a greater offset of their headings from the windflow in an attempt to partially correct for the drift.

Finally, we investigated the impact that seasonal differences in wind selectivity and orientation performance had on the displacement speed of the migrant hoverflies. The mean wind speeds during mass migrations were relatively low (spring: 6.6 m s$^{-1}$; autumn: 5.8 m s$^{-1}$), with 50% of occasions with

speeds between 4 and 9 m s$^{-1}$ (figure 4a), and they were significantly slower than winds during non-mass migration days (spring: 8.5 m s$^{-1}$; autumn: 8.6 m s$^{-1}$; Scheirer–Ray–Hare test: $H = 110$, $p < 0.001$). Winds were slightly faster in spring than autumn (9.27 versus 9.19 m s$^{-1}$; electronic supplementary material, figure S7); however, the effect of season on wind speed was not significant ($H = 0.25$, $p = 0.620$), although there was a significant interaction between season and migration ($H = 4.9$, $p = 0.027$), indicating that hoverflies selected weather conditions with the slowest winds during autumn mass migrations (figure 4a). Displacement speeds were somewhat faster than wind speeds, signifying that much of the self-powered flight vector was directed in the downstream direction. However, the combination of selection of slightly slower winds, plus a greater offset of the heading vector from the downwind owing to correction for lateral drift, resulted in significantly slower displacement speeds during the autumn than the spring (spring: 11.2 m s$^{-1}$; autumn: 9.8 m s$^{-1}$; $t$-test: $t_{382} = 5.05$, $p < 0.0001$; figure 4b).

## 4. Discussion

In this study, we demonstrate that high-flying diurnal migrant hoverflies, using fast-moving airstreams at heights greater than 150 m above the ground, use a combination of flight behaviours and orientation responses to achieve rapid, long-range movement in seasonally beneficial directions. Similar strategies have been studied in large *nocturnal* migrants at high altitude [14,15,32]; however, this is, to our knowledge, the first demonstration of the existence of such strategies in day-flying meso-insects, which undertake bidirectional seasonal migrations in enormous numbers [3]. Migrant hoverflies are a major constituent of this category of insect migrant [5], and their movements are of great consequence owing to the provision of key ecosystem services, and so an understanding of their migration strategies is important.

During the spring, migrant hoverflies arrive in Britain on southerly winds that aid their transport in a seasonally beneficial direction. This pattern may simply result from the fact that source populations of migrants are only available when winds blow from further south, owing to the geographical distribution of migrant hoverflies at this time of year [5]. Because of the strong positive effect of warmer air temperatures on the intensity of insect migration [3], the suitability of such winds is enhanced by the fact that southerly winds tend to be warmer than those from other directions. Thus, there may be an absence of wind selectivity during the spring, with the pattern observed merely a consequence of the geographical distribution of migrants at this time and/or the positive influence of warmer winds. If this were the case, we might expect spring migrants to show no particular orientation strategy, but to fly more-or-less downwind in order to increase their displacement speed. This appears to be the case, as spring migrants exhibited markedly poorer orientation performance than autumn migrants, and they did not attempt to correct for wind drift away from the seasonal PDM (0°). The strategy in the spring, therefore, appears to be one of aligning their heading vector relatively close to the downwind direction (albeit with a fair degree of error), but without any evidence for compensatory orientation. When this is combined with the positive influence of warmer winds on the probability of initiating migration, this strategy should be sufficient for a general northward expansion [9].

By contrast, autumn migrants employed a combination of migration decisions and orientation responses that ensured displacement towards the autumn PDM (198°), despite the unfavourableness of the wind. The first stage is that mass migration only occurs on days on which the wind direction is more favourable than the average situation (i.e. with a greater southward component). They also showed a tendency to migrate on days with slower winds than the average, and although this was true in both seasons, the effect was most pronounced during the autumn. The selection of relatively slow winds, blowing closer to the seasonal PDM, provides some benefit to the autumn migrants, but on its own is not enough to allow them to migrate towards the south. In order to achieve this, the autumn migrants also showed a strong pattern of collective orientation towards their seasonal PDM. Our estimate for the autumn PDM (198°) is very similar to previous observations of flight directions of hoverflies migrating close to the ground across western Europe (see fig. 1 in Odermatt *et al.* [33]). Insects migrating close to the ground have greater control over their flight direction than high-flying migrants, and thus, we assume their ground direction is likely to be similar to their PDM. The similarity between the field observations and our estimate therefore strengthens our conviction that the autumn PDM is actually towards 198° rather than due south. This autumn migration direction would appear to be beneficial in western Europe as it follows the coastline towards the Mediterranean, and similar headings are frequently observed in studies of insect and bird migration [15,27]. The orientation strategy hoverflies employ to migrate towards the PDM, known as CBDO, involves offsetting the heading by only a small degree from the downwind direction [26], and only when the difference between the downwind and the PDM increases beyond some threshold (figure 3). Employment of the CBDO strategy maximizes the speed of travel (because the majority of the self-powered flight vector is aligned with the flow), but also allows for relatively modest (but significant) influence on the migration direction resulting in travel closer to the PDM than flying downwind [27]. This strategy suits short-lived migrants such as insects, where speed and distance travelled is of prime importance, but migration to a highly specific goal is not important. It is clear from our results that migrant hoverflies do not fit the idea of aerial plankton, but rather that they are actively orienting migrants using favourable winds for bidirectional migration.

Our results indicate that migrant hoverflies must have: (i) an internal compass that allows them to identify their seasonal PDM, and (ii) a mechanism for identifying the wind direction. The nature of the compass mechanism has not been elucidated in hoverflies as yet. Another dipteran (*Drosophila*) is able to maintain a fixed course along an arbitrary (but individually consistent) direction by visually navigating with respect to both celestial patterns of polarized light [34] and the position of the Sun [35]. Given the conserved nature of the 'behavioural toolkit' involved in movement [36], it seems likely that hoverflies would have the same visual capacity and thus, the ability to maintain a fixed (albeit arbitrarily directed) course with respect to either of these cues. A solar compass is a well-known feature of butterfly migration [17,37], and thus we assume that a time-compensated sun compass (based on the position of the Sun, polarized skylight and/or the chromatic or intensity gradient of the sky) will prove to be the mechanism used by migrant hoverflies to migrate in

seasonally beneficial directions. Our results showing that directional patterns remain stable over the course of the day (see the electronic supplementary material, figure S6 and table S4), despite the apparent motion of the Sun throughout the day, is consistent with the use of a fully time-compensated sun compass. The identification of the wind direction is relatively simple for diurnal migrants, as wind direction at ground level is very similar to direction at altitude [3], and so before migratory take-off the wind direction can be compared with the preferred direction.

Greater uncertainty surrounds the marked seasonal difference in orientation performance. It has been previously predicted that in species which seasonally invade higher latitudes, natural selection for mechanisms which enable migrants to select favourable winds and orient appropriately will be much stronger for autumn emigrants escaping deteriorating conditions than the original spring immigrants [9]. Indeed, this pattern of stronger orientation performance in the autumn has been seen in previous studies of insect migration [27,38] as well as in the migrant hoverflies studied here. However, whether this is a result of a greater motivation to migrate in a specific direction in the autumn migrants, or an increased capability to detect, and/or maintain flight along the preferred direction, all remain to be discovered.

Migrant hoverflies are just one component of the highly diverse and hugely abundant meso-insect community engaging in seasonally directed diurnal migrations above the UK [3,5], and in even larger numbers in warmer regions [12,13]. The hoverfly migration strategies are therefore likely to be widespread in this category of abundant diurnal insect migrants, which in the UK are mostly beneficial species [3,5]. In other regions of the world, this size category may also include serious pests of agriculture, livestock production and/or human health, and knowledge of the predictability of their migratory directions will be key to benefiting from, or coping with, the long-distance movements of these species. Thus, further investigation of this understudied group of important migrants is warranted.

Data accessibility. The data used in this manuscript have been deposited in the Dryad Digital Repository: https://doi.org/10.5061/dryad.w9ghx3fm1 [39].

Authors' contributions. J.W.C., B.G., G.H., B.-P.Z. and K.R.W. designed the study; B.G. carried out the analyses; M.H.M.M. and D.R.R. provided hoverfly and migration expertise; W.L.S.H. analysed hoverfly flight angles; B.G. and J.W.C. wrote the paper, with input from all authors.

Competing interests. We declare we have no competing interests.

Funding. This work was supported through grants to G.H. by the National Natural Science Foundation of China (grant no. 31822043), the Natural Science Foundation of Jiangsu Province (grant no. BK20170026) and the Postgraduate Research & Practice Innovation Program of Jiangsu Province (grant no. KYCX19_0533). K.R.W. was supported by the Royal Society University Research Fellowship scheme (grant no. UF150126) and a Research Grants for Research Fellows (grant no. RGF\R1\180047) that also supported the studentship of W.L.S.H. M.H.M.M. received funding from the European Union's Horizon 2020 research and innovation programme under the Marie Skłodowska-Curie Grant Agreement No. 795568 (InsectMigration). Rothamsted Research receives grant-aided support from the United Kingdom Biotechnology and Biological Sciences Research Council (BBSRC). B.G.'s visiting scholarship to the University of Exeter was funded by the China Scholarship Council.

Acknowledgements. Dr Özge Özden provided logistical support for the fieldwork in Cyprus.

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
