## [Reviewer comments · Proceedings of the Royal Society B: Biological Sciences]

Review History

RSPB-2019-2299.R0 (Original submission)

Review form: Reviewer 1

Recommendation

Major revision is needed (please make suggestions in comments)

Scientific importance: Is the manuscript an original and important contribution to its field?

Good

General interest: Is the paper of sufficient general interest?

Good

Quality of the paper: Is the overall quality of the paper suitable?

Marginal

Is the length of the paper justified?

Yes

Should the paper be seen by a specialist statistical reviewer?

Yes

Do you have any concerns about statistical analyses in this paper? If so, please specify them explicitly in your report.

Yes

It is a condition of publication that authors make their supporting data, code and materials available - either as supplementary material or hosted in an external repository. Please rate, if applicable, the supporting data on the following criteria.

Is it accessible?

No

Is it clear?

No

Is it adequate?

No

Do you have any ethical concerns with this paper?

No

Comments to the Author

This is a really interesting paper on hoverfly migrations in England, and their capacity to use favorable winds and compensate for drift. The small size of hoverflies makes them very difficult to track, but the authors have found ways to distinguish their mass and body shapes from the other objects amassed in the data collected by vertical looking radars. Their small size and slow airspeeds make them particularly vulnerable to the winds. The authors conclude that the insects drift with the winds in the spring when the prevailing winds and their general direction of the hoverfly migration are the same; whereas the insects compensate to some degree for drift in the autumn, when the prevailing winds are contrary to the migratory direction.

My concern with this paper is the purpose is not well defined. It appears that they wish to identify the 'precise mechanisms by which hoverflies achieve movement along their seasonal' preferred direction of movement (PDM, l. 115). A compass is introduced in l. 108, and so is the selection of tailwinds (l. 113). It is unclear whether both of these two mechanisms for maintaining a migratory direction are objectives of this paper. A third objective that does seem clear is to refute the moniker that meso-insects like hoverflies are 'aerial plankton' (l. 51, l. 114), passively drifting on the winds. However because it is stated in the Introduction that hoverflies are 'completely at odds with the assumption of passively drifting aerial plankton' (l. 114), it appears that this has been shown previously. A bit of reorganization of the Introduction, Results, and less repetition between the Introduction and the Discussion might help to clarify what is known beforehand, the purpose of the paper, the hypotheses tested, and what the new data reveal. Presenting results from a previous paper in the Results section is extremely odd and confusing (l. 250-276). These should be incorporated into the Introduction or perhaps a separate section on Study Organisms.

More general comments:

Flight data from three locations are pooled without testing whether the orientations of insects and winds are significantly different first. Insect orientations might differ among locations.

The methods for calculating drift are very difficult to follow and would be aided by a supplemental figure or two.

The authors also incorporate several assumptions into the methods that might alter the outcome if the assumptions proved false. For example, the authors assume a migration towards due North in the spring and towards due South in the autumn. An analysis of heading on crosswind

drift should yield the PDM for each season without having to assume it. In fact, the authors carry out such an analysis (Fig. 4), but do not use this value but instead use the assumed PDM in their method. If they did the analysis with the calculated value, which was significantly different from due South, would it change their results?

The authors state that the prevailing winds are the same in autumn and spring, but examination of figure 1 suggests that the prevailing wind is from the southwest in spring and more westerly in autumn. Could they test their assumption that the winds are consistently from the southwest (l. 104) at the sites of the vertical looking radar?

The authors take directions of multiple individuals and combine them into a mean track and heading direction for each migratory day. However the mean orientations have not only a direction to the vector but also a length that varies with concentration, and so when analyzing the mean of these means, second order circular statistics should be applied (see Zar or Batschelet for methods). This may also be true for the wind measurements, but it is unclear how many directions are collected each day.

In figure 4, the values on the y-axis are compass degrees, which are not linear. Rather they are circular, and the authors need to apply circular statistics to these data rather than the linear regressions that they used. Alternatively, linear regression can be applied if the compass directions are limited to a semi-circle (180° , see Zar).

Specific comments:

l. 327 although the potential for modest partial compensation has not been eliminated by the CI (0.75, 1.12).

l. 344 There is no difference in the winds on non-migratory days but winds are significantly slower on autumn migratory days. Doesn't there have to be a different distribution of winds from which to select for this to be true? Is the distribution of wind speeds in spring and autumn the same?

l. 377 If they were simply flying downwind, then axis of orientation would equal wind direction.

l. 382 'Restriction of behavior' may be too extreme a term. They may actually attempt to migrate but find it to no avail and drop out. As a result, migrating insects are rare and observed less or excluded as too few to be deemed a mass migration in unfavorable winds.

l. 412 Although a compass might provide a general direction, it may not need to be time compensated. Leading lines are a real possibility for maintaining a common orientation throughout the daytime.

Review form: Reviewer 2

Recommendation

Major revision is needed (please make suggestions in comments)

Scientific importance: Is the manuscript an original and important contribution to its field?

Excellent

General interest: Is the paper of sufficient general interest?

Good

Quality of the paper: Is the overall quality of the paper suitable?

Good

Is the length of the paper justified?

Yes

Should the paper be seen by a specialist statistical reviewer?

No

Do you have any concerns about statistical analyses in this paper? If so, please specify them explicitly in your report.

Yes

It is a condition of publication that authors make their supporting data, code and materials available - either as supplementary material or hosted in an external repository. Please rate, if applicable, the supporting data on the following criteria.

Is it accessible?

Yes

Is it clear?

Yes

Is it adequate?

Yes

Do you have any ethical concerns with this paper?

No

Comments to the Author

In this manuscript, the authors present data from vertical-looking entomological radar, gleaned flight velocities (tracks) and body orientations (headings) of individual hoverflies. These flies are known to conduct true latitudinal migrations, flying south in autumn and north in spring. This manuscript addresses how this is achieved, given that prevailing winds blow to the north year-round, and are generally much faster than the top airspeeds these hoverflies can achieve. First, they show that their radar is far more likely to detect these flies aloft on days with winds that are: 1) more aligned with their preferred direction of migration (PDM) than are the prevailing winds, and 2) lower than average speed. Both these trends are stronger in the autumn datasets (when winds were often opposing the southbound migration) and are taken as evidence that hoverflies elect to migrate on favorable winds. The authors then go on to investigate the hoverflies' flight behaviors in the context of high-altitude winds, calculating individual flies' heading offsets, that is the angular difference (α) between track and heading. They show that, in the autumn but not the spring, flies maintain azimuthal headings that are significantly closer to their PDM than were their tracks; this orientation behavior was driven by migration events whose mean drift angles (the offset between their actual track and their PDM) were particularly large. Finally, by subjecting flies' headings and tracks to an established regression analysis, the authors suggest that hoverfly populations migrating in autumn employ a fixed heading strategy, rather than adjusting their body orientation to achieve fixed trajectories. This study certainly advances our understanding of insect migration in the wild and is well worth publishing; however, the manuscript would be improved by considering the major and minor comments listed below.

Major concerns

1) In the abstract (line 34) and discussion (lines 411 – 417), the authors assume hoverflies employ time compensation. However, I think it should be possible to test this idea more explicitly using the data at hand - an exercise that would improve the depth and impact of the paper. In particular, it should be possible to parse the autumn mass-migration events into a.m. and p.m.

epochs (based on the time course of the raw VLR observations - not the daily means) and then regress the average tracks on the average alpha value within the two datasets. If these hoverflies employ time compensation, these two regression lines should have very similar intercepts. On the other hand, if hoverflies instead maintain fixed (uncompensated) headings with respect to the moving sun, one would expect shifted intercepts for a.m. versus p.m. data (see drawing at right).

It may also be worth noting the hoverflies might not appreciably benefit from the increased control over heading provided by time compensation. Figure 1 in Guilford and Taylor's 2014 paper provides an illustration of theoretical headings adopted, over the course of a day, by a migrant employing celestial navigation with and without time compensation. Their figure suggests situations in which a migrant lacking a time-compensation mechanism may yet be able to maintain an acceptably well-regulated heading; for example, if the migrants only fly for a few hours on any given day, the angular deviation of their self-powered trajectory would be modest.

2) I have some concerns with the fact that, in all analyses, the radar data were first compressed into daily means (mean tracks, mean headings) across individual hoverflies. I understand that regressing track angles on α is a powerful way to assess migration strategies, given that this method capitalizes on the variability in wind conditions among mass-migration events, and does not require the destination [PDM] to be specified a priori (Green and Alerstam 2002). From reading Green and Alerstam's work, I understand that a regression analysis performed on individual migrant data can yield spurious results if the dataset does not derive from a uniform circular distribution of wind directions. Specifically, their modeling indicates that datasets dominated by a broad range of tailwinds (a scenario obviously applying to the manuscript at hand) are prone to falsely report a behavioral strategy of overcompensation. I assume this explains why the authors have decided to report means for each mass migration event. If this is true, I think it would help future readers to say something more explicit about this justification. Until digging extensively into Green and Alerstam (2002), I was perplexed as to why the paper deals only with daily means.

That said, I still think that some possibly important information is lost by only considering daily means. For example, when first looking at Figure 3, I wanted to know which scatter points represented a mass migration of, say, many hundreds of individual hoverflies, and which represented far fewer observations. For Figure 3b specifically, I wondered how the regression slope might change if the fitting procedure was weighted to account for the number of raw observations underlying each point. Might outlier points represent daily migrations with comparatively few flies? By squinting at Figure 3a, one can see a possible underlying upward trend in the data, which might be significant if the outlier points did indeed represent situations based on relatively few individuals. In any event, I think the authors would be well served by addressing this issue.

I also wonder why daily means were used Figures 1 and 2. Why not simply present circular histograms of all individual data, pooled over mass-migration events, for spring and autumn? The authors could consider using a ranked Rayleigh test (Moore 1980) to account for varying vector lengths of each mass-migration event, but to my understanding this does not explicitly deal with variable sample sizes in each daily mean. Would be more straightforward to run a single Rayleigh test on all individual data? As mentioned above, analyzing individual observations (and when they were made throughout the day) might provide a means of testing for time compensation.

3) I think the filtering scheme used to select for hoverflies (selecting a range of body masses and reflectivity shape value ratios) is insufficiently described in this paper, given its importance for all subsequent analyses. I do appreciate that this procedure is clearly grounded on prior work (Wotton 2019). However, given the importance this supporting information ought to be concisely summarized in the methods section of this present manuscript as well. Furthermore, I think it would be helpful for the authors to report pre-filtered versus post-filtered insect counts, for both "mass-migration" and "non-mass-migration" events. Knowing what percentage of all individual

observations were classified as hoverflies (and what percentage were not) would be very helpful in interpreting both the methodology and the underlying natural phenomena. This should be provided in the text.

A related point concerns the question of whether a hoverfly's body pitch influences the $\sigma_{xx} / \sigma_{yy}$ ratio generated by its body (and I certainly defer to the authors' expertise with radar data). Could this ratio vary as the cosine of the insects' body pitch relative to the horizontal? I was curious about this point because there is ample evidence that flies alter body pitch as one mechanism to modulate forward airspeed – which might be relevant to data collected under different wind conditions. Is there any chance that observations might be mis-classified (either false negatives or positives) depending on the body pitch of the insect?

4) It would be helpful to present, in or alongside Figure 3b, mean headings also regressed on mean alpha values. My understanding of Green and Alerstam's regression method for assessing drift strategies (2002) is that a full drift (or constant heading) strategy is characterized by a track/alpha slope of one, and a heading/alpha slope of zero. This manuscript presents evidence in support of the former, but seemingly not of the latter.

Minor comments

Figure 2: The sub-analyses in Figure 2 (b) and (d) are mostly lucid, but one aspect of these circular histograms is confusing: the overall skylines of the histograms (panel a versus panel b, and panel c versus panel d) appear to have been altered by the process of fractioning the datapoints into those with track $> 20^\circ$ from PDM and track $< 20^\circ$ from PDM. I am guessing that the histogram binning was somehow perturbed by this layer of analysis and think addressing this would improve the figure's clarity.

Figure 3: The analysis in Figure 2 was conducted with respect to an assumed autumn PDM of 180° , but Figure 3's regression analysis revealed the PDM to be 189° . Does the finding from Figure 2 change appreciably if the updated estimate of PDM is used?

L36: "...arriving in the UK in spring, showed less orientation ability..." I suggest phrasing this more phenomenologically, because its current wording raises some questions about orientation mechanisms that the paper does not go on to address. Perhaps "... showed weaker orientation tendencies" or even "... showed a lesser degree of orientation".

L316: "This result indicates that as the degree of ... drift away from the seasonal PDM ... increased, that hoverflies showed a greater offset..."
Omit one of the two "that's" in this sentence.

L320: "... assigned a positive or negative sign depending on whether the heading was to the left or right of the track..." I found the left/right designation confusing for circularly distributed data.

L323: "... spring hoverflies make no attempt to correct for drift." and L328: "[in autumn hoverflies keep a] 'constant compass course' irrespective of the wind"
The distinction between these two inferred behavioral strategies ought to be made clearer by editing line 323. At first glance "making no attempt to correct for drift" and "full drift" sound very similar, but the latter refers to a distinct strategy in which mean heading is fixed and wind drift fully interacts with the fixed heading to produce the track.

L339: "...95% of occasions with speeds between 4 and 9 m s⁻¹ (figure 4a)..."
The description of the box/whisker plots in Figure legend 4a suggests that actually only 50% (the interquartile range) of mass-migration occasions had wind speeds between 4 and 9 m s⁻¹.

L547 (figure legends): "... mass migration or a non-migration event."

I think the authors meant to write "non-mass-migration event."

Decision letter (RSPB-2019-2299.R0)

03-Dec-2019

Dear Dr Chapman:

I am writing to inform you that your manuscript RSPB-2019-2299 entitled "Adaptive strategies of high-flying migratory hoverflies in response to wind currents" has, in its current form, been rejected for publication in Proceedings B.

This action has been taken on the advice of referees, who have recommended that substantial revisions are necessary. With this in mind we would be happy to consider a resubmission, provided the comments of the referees are fully addressed. However please note that this is not a provisional acceptance.

Please note that this decision may (or may not) have taken into account confidential comments.

In your revision process, please take a second look at how open your science is; our policy is that all data involved with the study should be made openly accessible-- see: <https://royalsociety.org/journals/ethics-policies/data-sharing-mining/>
Insufficient sharing of data can delay or even cause rejection of a paper.

Sincerely,
Professor John Hutchinson, Editor
mailto: proceedingsb@royalsociety.org

Comments to Author:
Associate Editor: Doug Altshuler

The authors have applied radar measurements to investigate the migrations of hoverflies, given

that their flight speeds are typically lower than prevailing wind speeds. It is a fascinating topic and this study provide novel insight. We have now obtained two expert reviews, and both are enthusiastic about the study. Both referees also have significant concerns with the data analysis and presentation. I encourage the authors to revise the manuscript accordingly because addressing these concerns will greatly strengthen the contribution.

Reviewer(s)' Comments to Author:

Referee: 1

Comments to the Author(s)

This is a really interesting paper on hoverfly migrations in England, and their capacity to use favorable winds and compensate for drift. The small size of hoverflies makes them very difficult to track, but the authors have found ways to distinguish their mass and body shapes from the other objects amassed in the data collected by vertical looking radars. Their small size and slow airspeeds make them particularly vulnerable to the winds. The authors conclude that the insects drift with the winds in the spring when the prevailing winds and their general direction of the hoverfly migration are the same; whereas the insects compensate to some degree for drift in the autumn, when the prevailing winds are contrary to the migratory direction.

My concern with this paper is the purpose is not well defined. It appears that they wish to identify the 'precise mechanisms by which hoverflies achieve movement along their seasonal' preferred direction of movement (PDM, l. 115). A compass is introduced in l. 108, and so is the selection of tailwinds (l. 113). It is unclear whether both of these two mechanisms for maintaining a migratory direction are objectives of this paper. A third objective that does seem clear is to refute the moniker that meso-insects like hoverflies are 'aerial plankton' (l. 51, l. 114), passively drifting on the winds. However because it is stated in the Introduction that hoverflies are 'completely at odds with the assumption of passively drifting aerial plankton' (l. 114), it appears that this has been shown previously. A bit of reorganization of the Introduction, Results, and less repetition between the Introduction and the Discussion might help to clarify what is known beforehand, the purpose of the paper, the hypotheses tested, and what the new data reveal. Presenting results from a previous paper in the Results section is extremely odd and confusing (l. 250-276). These should be incorporated into the Introduction or perhaps a separate section on Study Organisms.

More general comments:

Flight data from three locations are pooled without testing whether the orientations of insects and winds are significantly different first. Insect orientations might differ among locations.

The methods for calculating drift are very difficult to follow and would be aided by a supplemental figure or two.

The authors also incorporate several assumptions into the methods that might alter the outcome if the assumptions proved false. For example, the authors assume a migration towards due North in the spring and towards due South in the autumn. An analysis of heading on crosswind drift should yield the PDM for each season without having to assume it. In fact, the authors carry out such an analysis (Fig. 4), but do not use this value but instead use the assumed PDM in their method. If they did the analysis with the calculated value, which was significantly different from due South, would it change their results?

The authors state that the prevailing winds are the same in autumn and spring, but examination of figure 1 suggests that the prevailing wind is from the southwest in spring and more westerly in autumn. Could they test their assumption that the winds are consistently from the southwest (l. 104) at the sites of the vertical looking radar?

The authors take directions of multiple individuals and combine them into a mean track and heading direction for each migratory day. However the mean orientations have not only a direction to the vector but also a length that varies with concentration, and so when analyzing the mean of these means, second order circular statistics should be applied (see Zar or Batschelet for methods). This may also be true for the wind measurements, but it is unclear how many directions are collected each day.

In figure 4, the values on the y-axis are compass degrees, which are not linear. Rather they are circular, and the authors need to apply circular statistics to these data rather than the linear regressions that they used. Alternatively, linear regression can be applied if the compass directions are limited to a semi-circle (180°, see Zar).

Specific comments:

l. 327 although the potential for modest partial compensation has not been eliminated by the CI (0.75, 1.12).

l. 344 There is no difference in the winds on non-migratory days but winds are significantly slower on autumn migratory days. Doesn't there have to be a different distribution of winds from which to select for this to be true? Is the distribution of wind speeds in spring and autumn the same?

l. 377 If they were simply flying downwind, then axis of orientation would equal wind direction.

l. 382 'Restriction of behavior' may be too extreme a term. They may actually attempt to migrate but find it to no avail and drop out. As a result, migrating insects are rare and observed less or excluded as too few to be deemed a mass migration in unfavorable winds.

l. 412 Although a compass might provide a general direction, it may not need to be time compensated. Leading lines are a real possibility for maintaining a common orientation throughout the daytime.

Referee: 2

Comments to the Author(s)

In this manuscript, the authors present data from vertical-looking entomological radar, gleaned flight velocities (tracks) and body orientations (headings) of individual hoverflies. These flies are known to conduct true latitudinal migrations, flying south in autumn and north in spring. This manuscript addresses how this is achieved, given that prevailing winds blow to the north year-round, and are generally much faster than the top airspeeds these hoverflies can achieve. First, they show that their radar is far more likely to detect these flies aloft on days with winds that are: 1) more aligned with their preferred direction of migration (PDM) than are the prevailing winds, and 2) lower than average speed. Both these trends are stronger in the autumn datasets (when winds were often opposing the southbound migration) and are taken as evidence that hoverflies elect to migrate on favorable winds. The authors then go on to investigate the hoverflies' flight behaviors in the context of high-altitude winds, calculating individual flies' heading offsets, that is the angular difference (α) between track and heading. They show that, in the autumn but not the spring, flies maintain azimuthal headings that are significantly closer to their PDM than were their tracks; this orientation behavior was driven by migration events whose mean drift angles (the offset between their actual track and their PDM) were particularly large. Finally, by subjecting flies' headings and tracks to an established regression analysis, the authors suggest that hoverfly populations migrating in autumn employ a fixed heading strategy, rather than adjusting their body orientation to achieve fixed trajectories. This study certainly advances our

understanding of insect migration in the wild and is well worth publishing; however, the manuscript would be improved by considering the major and minor comments listed below.

Major concerns

1) In the abstract (line 34) and discussion (lines 411 – 417), the authors assume hoverflies employ time compensation. However, I think it should be possible to test this idea more explicitly using the data at hand - an exercise that would improve the depth and impact of the paper. In particular, it should be possible to parse the autumn mass-migration events into a.m. and p.m. epochs (based on the time course of the raw VLR observations - not the daily means) and then regress the average tracks on the average α value within the two datasets. If these hoverflies employ time compensation, these two regression lines should have very similar intercepts. On the other hand, if hoverflies instead maintain fixed (uncompensated) headings with respect to the moving sun, one would expect shifted intercepts for a.m. versus p.m. data (see drawing at right).

It may also be worth noting that hoverflies might not appreciably benefit from the increased control over heading provided by time compensation. Figure 1 in Guilford and Taylor's 2014 paper provides an illustration of theoretical headings adopted, over the course of a day, by a migrant employing celestial navigation with and without time compensation. Their figure suggests situations in which a migrant lacking a time-compensation mechanism may yet be able to maintain an acceptably well-regulated heading; for example, if the migrants only fly for a few hours on any given day, the angular deviation of their self-powered trajectory would be modest.

2) I have some concerns with the fact that, in all analyses, the radar data were first compressed into daily means (mean tracks, mean headings) across individual hoverflies. I understand that regressing track angles on α is a powerful way to assess migration strategies, given that this method capitalizes on the variability in wind conditions among mass-migration events, and does not require the destination [PDM] to be specified a priori (Green and Alerstam 2002). From reading Green and Alerstam's work, I understand that a regression analysis performed on individual migrant data can yield spurious results if the dataset does not derive from a uniform circular distribution of wind directions. Specifically, their modeling indicates that datasets dominated by a broad range of tailwinds (a scenario obviously applying to the manuscript at hand) are prone to falsely report a behavioral strategy of overcompensation. I assume this explains why the authors have decided to report means for each mass migration event. If this is true, I think it would help future readers to say something more explicit about this justification. Until digging extensively into Green and Alerstam (2002), I was perplexed as to why the paper deals only with daily means.

That said, I still think that some possibly important information is lost by only considering daily means. For example, when first looking at Figure 3, I wanted to know which scatter points represented a mass migration of, say, many hundreds of individual hoverflies, and which represented far fewer observations. For Figure 3b specifically, I wondered how the regression slope might change if the fitting procedure was weighted to account for the number of raw observations underlying each point. Might outlier points represent daily migrations with comparatively few flies? By squinting at Figure 3a, one can see a possible underlying upward trend in the data, which might be significant if the outlier points did indeed represent situations based on relatively few individuals. In any event, I think the authors would be well served by addressing this issue.

I also wonder why daily means were used in Figures 1 and 2. Why not simply present circular histograms of all individual data, pooled over mass-migration events, for spring and autumn? The authors could consider using a ranked Rayleigh test (Moore 1980) to account for varying vector lengths of each mass-migration event, but to my understanding this does not explicitly deal with variable sample sizes in each daily mean. Would be more straightforward to run a single Rayleigh test on all individual data? As mentioned above, analyzing individual observations (and when they were made throughout the day) might provide a means of testing for time compensation.

3) I think the filtering scheme used to select for hoverflies (selecting a range of body masses and reflectivity shape value ratios) is insufficiently described in this paper, given its importance for all subsequent analyses. I do appreciate that this procedure is clearly grounded on prior work (Wotton 2019). However, given the importance this supporting information ought to be concisely summarized in the methods section of this present manuscript as well. Furthermore, I think it would be helpful for the authors to report pre-filtered versus post-filtered insect counts, for both “mass-migration” and “non-mass-migration” events. Knowing what percentage of all individual observations were classified as hoverflies (and what percentage were not) would be very helpful in interpreting both the methodology and the underlying natural phenomena. This should be provided in the text.

A related point concerns the question of whether a hoverfly’s body pitch influences the $\sigma_{xx} / \sigma_{yy}$ ratio generated by its body (and I certainly defer to the authors’ expertise with radar data). Could this ratio vary as the cosine of the insects’ body pitch relative to the horizontal? I was curious about this point because there is ample evidence that flies alter body pitch as one mechanism to modulate forward airspeed – which might be relevant to data collected under different wind conditions. Is there any chance that observations might be mis-classified (either false negatives or positives) depending on the body pitch of the insect?

4) It would be helpful to present, in or alongside Figure 3b, mean headings also regressed on mean alpha values. My understanding of Green and Alerstam’s regression method for assessing drift strategies (2002) is that a full drift (or constant heading) strategy is characterized by a track/alpha slope of one, and a heading/alpha slope of zero. This manuscript presents evidence in support of the former, but seemingly not of the latter.

Minor comments

Figure 2: The sub-analyses in Figure 2 (b) and (d) are mostly lucid, but one aspect of these circular histograms is confusing: the overall skylines of the histograms (panel a versus panel b, and panel c versus panel d) appear to have been altered by the process of fractioning the datapoints into those with track $> 20^\circ$ from PDM and track $< 20^\circ$ from PDM. I am guessing that the histogram binning was somehow perturbed by this layer of analysis and think addressing this would improve the figure’s clarity.

Figure 3: The analysis in Figure 2 was conducted with respect to an assumed autumn PDM of 180° , but Figure 3’s regression analysis revealed the PDM to be 189° . Does the finding from Figure 2 change appreciably if the updated estimate of PDM is used?

L36: “...arriving in the UK in spring, showed less orientation ability...” I suggest phrasing this more phenomenologically, because its current wording raises some questions about orientation mechanisms that the paper does not go on to address. Perhaps “... showed weaker orientation tendencies” or even “... showed a lesser degree of orientation”.

L316: “This result indicates that as the degree of ... drift away from the seasonal PDM ... increased, that hoverflies showed a greater offset...”
Omit one of the two “that’s” in this sentence.

L320: “... assigned a positive or negative sign depending on whether the heading was to the left or right of the track...” I found the left/right designation confusing for circularly distributed data.

L323: “... spring hoverflies make no attempt to correct for drift.” and L328: “[in autumn hoverflies keep a] ‘constant compass course’ irrespective of the wind ”

The distinction between these two inferred behavioral strategies ought to be made clearer by editing line 323. At first glance “making no attempt to correct for drift” and “full drift” sound

very similar, but the latter refers to a distinct strategy in which mean heading is fixed and wind drift fully interacts with the fixed heading to produce the track.

L339: "...95% of occasions with speeds between 4 and 9 m s⁻¹ (figure 4a)..."

The description of the box/whisker plots in Figure legend 4a suggests that actually only 50% (the interquartile range) of mass-migration occasions had wind speeds between 4 and 9 m s⁻¹.

L547 (figure legends): "... mass migration or a non-migration event."

I think the authors meant to write "non-mass-migration event."

Author's Response to Decision Letter for (RSPB-2019-2299.R0)

See Appendix A.

RSPB-2020-0406.R0

Review form: Reviewer 1 (Robert Srygley)

Recommendation

Major revision is needed (please make suggestions in comments)

Scientific importance: Is the manuscript an original and important contribution to its field?

Excellent

General interest: Is the paper of sufficient general interest?

Excellent

Quality of the paper: Is the overall quality of the paper suitable?

Good

Is the length of the paper justified?

No

Should the paper be seen by a specialist statistical reviewer?

No

Do you have any concerns about statistical analyses in this paper? If so, please specify them explicitly in your report.

No

It is a condition of publication that authors make their supporting data, code and materials available - either as supplementary material or hosted in an external repository. Please rate, if applicable, the supporting data on the following criteria.

Is it accessible?

No

Is it clear?

N/A

Is it adequate?

N/A

Do you have any ethical concerns with this paper?

No

Comments to the Author

Review for Revision 1 of Gao et al. Adaptive strategies for high-flying migratory hoverflies...

I have reviewed this paper before and I appreciate the authors attention to improving the manuscript. My principal suggestion is a more detailed analysis of time compensation, why I detail in the minor comments below. There is still a lot of repetition between the Introduction and Discussion which would shorten the Discussion if revised. In addition, it remains unclear what data has been published in the previous paper on the topic. Is it all of Figure 1, or Figure 1a?

l. 101 change 'migrate' to 'fly' since migrate repeats migrant

l. 108 Lines 276-281 on directions of migrants should be moved to this paragraph which provides background information

l. 252 change 'they were' to 'it was'

l. 266 state how many records per day

l. 269 change 'was' to 'were'

l. 281 It remains completely unclear what is new and what is previously published. By moving published material to the Introduction.

l. 284 References to an 8-point compass are very confusing. 84 degrees is actually closer to ENE than 61 degrees. Why not just state the degrees? Or refer to the first as 'more northeasterly' and the second one as 'more easterly'.

l. 290 In the text you compare the orientations to the total data set, but the statistic compares the two orientations. The text and the statistical test should agree. Which comparison is more important?

l. 292 I thought it was ENE

l. 337 Move the 95% C.I. to just after slope of 1.18 and delete 'do not overlap'

l. 338 If 1 is full compensation and 0 is no compensation, then greater than 1 is 'overcompensation' for wind drift, which means the insects are turning further into the wind than necessary to compensate for wind drift. It's unclear how you can call this 'compass-biased downstream orientation.'

l. 342 There are several ways to compensate for wind drift. Separation of the data into morning and afternoon can not distinguish between these mechanisms. For example, compensation in the morning and afternoon can be achieved with a stepfunction time compensation mechanism such as that possessed by naïve honeybees. Time averaging and full compensation can be investigated with hourly averages. See Oliveira et al. 1998 JeB or the chapter by Srygley and Oliveira in Insect Movement (2001, Fig. 9.3) for the method.

l. 346 Shouldn't 'autumn' be 'afternoon' in this sentence?

- l. 384 The Discussion stills seems repetitive of the Introduction.
- l. 428 The following statement is not true: Insect migrating close to the ground have complete control over their flight direction, and thus their ground direction can be assumed to be the same as their PDM. There are instances when drift would be the optimal strategy, and in those instances ground direction is not the same as the PDM.
- l. 438 'Employment'
- l. 460 a time-compensated celestial compass is not required. Only a global orientation cue. So magnetic compass would suffice. In this sentence, you really just want to focus on the time-compensation part as the 'celestial compass' has already been discussed in l. 456.
- l. 466 Change 'an' to 'and' You've only shown limited time-compensation. You should do the complete analysis that I suggest.
- l. 471 Delete second 'that'
- l. 478 Delete ', ' after along

Decision letter (RSPB-2020-0406.R0)

19-Mar-2020

Dear Dr Chapman:

Your manuscript has now been peer reviewed and the reviews have been assessed by an Associate Editor. The reviewers' comments (not including confidential comments to the Editor) and the comments from the Associate Editor are included at the end of this email for your reference. As you will see, the reviewers and the Editors have raised some concerns with your manuscript and we would like to invite you to revise your manuscript to address them.

When revising your manuscript you should also ensure that it adheres to our editorial policies

(<https://royalsociety.org/journals/ethics-policies/>). You should pay particular attention to the following:

Research ethics:

Use of animals and field studies:

Please submit a copy of your revised paper within three weeks. If we do not hear from you within this time your manuscript will be rejected. If you are unable to meet this deadline please let us know as soon as possible, as we may be able to grant a short extension.

Best wishes,
Dr John Hutchinson, Editor
mailto: proceedingsb@royalsociety.org

Associate Editor Board Member
Comments to Author:
Associate Editor: Doug Altshuler

The authors have submitted a revised and improved manuscript. One of the referees was unavailable. The other is generally positive about the manuscript but has some remaining major concerns. The main issue from my perspective pertains to the analysis of time compensation, and I would appreciate seeing if the authors can address this.

Reviewer(s)' Comments to Author:

Referee: 1

Comments to the Author(s).
Review for Revision 1 of Gao et al. Adaptive strategies for high-flying migratory hoverflies...

I have reviewed this paper before and I appreciate the authors attention to improving the manuscript. My principal suggestion is a more detailed analysis of time compensation, why I detail in the minor comments below. There is still a lot of repetition between the Introduction and Discussion which would shorten the Discussion if revised. In addition, it remains unclear what data has been published in the previous paper on the topic. Is it all of Figure 1, or Figure 1a?

1. 101 change 'migrate' to 'fly' since migrate repeats migrant
1. 108 Lines 276-281 on directions of migrants should be moved to this paragraph which provides background information
1. 252 change 'they were' to 'it was'
1. 266 state how many records per day
1. 269 change 'was' to 'were'
1. 281 It remains completely unclear what is new and what is previously published. By moving published material to the Introduction.
1. 284 References to an 8-point compass are very confusing. 84 degrees is actually closer to ENE than 61 degrees. Why not just state the degrees? Or refer to the first as 'more northeasterly' and the second one as 'more easterly'.
1. 290 In the text you compare the orientations to the total data set, but the statistic compares the two orientations. The text and the statistical test should agree. Which comparison is more important?
1. 292 I thought it was ENE
1. 337 Move the 95% C.I. to just after slope of 1.18 and delete 'do not overlap'

l. 338 If 1 is full compensation and 0 is no compensation, then greater than 1 is 'overcompensation' for wind drift, which means the insects are turning further into the wind than necessary to compensate for wind drift. It's unclear how you can call this 'compass-biased downstream orientation.'

l. 342 There are several ways to compensate for wind drift. Separation of the data into morning and afternoon can not distinguish between these mechanisms. For example, compensation in the morning and afternoon can be achieved with a stepfunction time compensation mechanism such as that possessed by naïve honeybees. Time averaging and full compensation can be investigated with hourly averages. See Oliveira et al. 1998 JeB or the chapter by Srygley and Oliveira in *Insect Movement* (2001, Fig. 9.3) for the method.

l. 346 Shouldn't 'autumn' be 'afternoon' in this sentence?

l. 384 The Discussion stills seems repetitive of the Introduction.

l. 428 The following statement is not true: Insect migrating close to the ground have complete control over their flight direction, and thus their ground direction can be assumed to be the same as their PDM. There are instances when drift would be the optimal strategy, and in those instances ground direction is not the same as the PDM.

l. 438 'Employment'

l. 460 a time-compensated celestial compass is not required. Only a global orientation cue. So magnetic compass would suffice. In this sentence, you really just want to focus on the time-compensation part as the 'celestial compass' has already been discussed in l. 456.

l. 466 Change 'an' to 'and' You've only shown limited time-compensation. You should do the complete analysis that I suggest.

l. 471 Delete second 'that'

l. 478 Delete ',' after along

Author's Response to Decision Letter for (RSPB-2020-0406.R0)

See Appendix B.

Decision letter (RSPB-2020-0406.R1)

04-May-2020

Dear Dr Chapman

I am pleased to inform you that your manuscript entitled "Adaptive strategies of high-flying migratory hoverflies in response to wind currents" has been accepted for publication in *Proceedings B*. Congratulations!!

Open Access

Paper charges

Sincerely,

Dr John Hutchinson

Associate Editor:

Board Member

Comments to Author:

Associate Editor: Doug Altshuler

The authors have done a fine job addressing the remaining concerns. Congratulations on an interesting study.

Appendix A

19 February 2020

Dear Editors of *Proc Roy Soc B*,

We wish to thank the reviewers for the extremely thorough and helpful review of our paper manuscript RSPB-2019-2299 ("Adaptive strategies of high-flying migratory hoverflies in response to wind currents). We have now revised our paper in line with all the comments from the referees, which we believe has substantially improved our study. In the letter below we provide a detailed point-by-point response to each of the referee's comments, and the areas we have changed in the text of our manuscript are highlighted in red font.

Referee: 1

Comments to the Author(s)

My concern with this paper is the purpose is not well defined. It appears that they wish to identify the 'precise mechanisms by which hoverflies achieve movement along their seasonal' preferred direction of movement (PDM, l. 115). A compass is introduced in l. 108, and so is the selection of tailwinds (l. 113). It is unclear whether both of these two mechanisms for maintaining a migratory direction are objectives of this paper. A third objective that does seem clear is to refute the moniker that meso-insects like hoverflies are 'aerial plankton' (l. 51, l. 114), passively drifting on the winds. However because it is stated in the Introduction that hoverflies are 'completely at odds with the assumption of passively drifting aerial plankton' (l. 114), it appears that this has been shown previously. A bit of reorganization of the Introduction, Results, and less repetition between the Introduction and the Discussion might help to clarify what is known beforehand, the purpose of the paper, the hypotheses tested, and what the new data reveal.

*** **Our Response:** We thank the referee for these useful comments, and have now reorganised the final section of the introduction (please see lines 100-116) to make it much clearer what had been previously discovered, and what was being specifically tested in this paper. To help with this, we have included a list of predictions we aim to test in the current paper.

Presenting results from a previous paper in the Results section is extremely odd and confusing (l. 250-276). These should be incorporated into the Introduction or perhaps a separate section on Study Organisms.

*** **Our Response:** In fact, the only repeated result from our previous hoverfly paper (ref 5) is the value of migratory direction in spring (342°) and autumn (180°), which we include on lines 277-280. The rest of the results in the paragraph the referee refers to (from lines 280 onwards) are completely new to this paper. We therefore feel it is appropriate to include the migration track directions in the results section, as they are crucial to understanding all the results which follow. We therefore prefer to leave the text as it is.

More general comments:

Flight data from three locations are pooled without testing whether the orientations of insects and winds are significantly different first. Insect orientations might differ among locations.

*** **Our Response:** We have now calculated and plotted all directional patterns (track, heading and downwind directions) for each of the three locations separately (see new Tables S1 & S2, and Figs S2

& S3). We tested these distributions using the Watson-Wheeler test, and in the majority of cases there were no significant differences between the directional data across the three sites (see new Table S3), and even where the distributions were significantly different, the mean direction were still rather similar. We have therefore decided it is reasonable to pool data across the three sites. We have included this new information in the Methods at lines 215-221.

The methods for calculating drift are very difficult to follow and would be aided by a supplemental figure or two.

***** Our Response:** We have produced a supplemental figure (see new Fig S4) which we feel will make our methods for calculating the drift angles much clearer, and we have added reference to the new figure in the methods section at lines 228, 251, 260, and in the legend of figure 3.

The authors also incorporate several assumptions into the methods that might alter the outcome if the assumptions proved false. For example, the authors assume a migration towards due North in the spring and towards due South in the autumn. An analysis of heading on crosswind drift should yield the PDM for each season without having to assume it. In fact, the authors carry out such an analysis (Fig. 3), but do not use this value but instead use the assumed PDM in their method. If they did the analysis with the calculated value, which was significantly different from due South, would it change their results?

***** Our Response:** We agree that this is a very valid point, and have indeed used the values from the regression analyses to determine the PDM. Based on responses to other points raised by Referees 1 and 2 (see below), we have also modified the linear regression method (by restricting to the linear section of the data, and using the least squares method – see response to Referee 2 below). Nothing materially changed for the spring dataset, and so we continue to use a presumed PDM of 0°.

However, in the autumn, the new value of the PDM calculated by the regression method was 198° (with 95% CI of 193-202°), which was clearly different from 180°. We therefore used a value of 198° for the PDM in the circular analysis of the degree of correction for drift. We have modified the text in the methods at lines 235-242, and the results at lines 339-342 to reflect this change. Because we use the PDM from the regression slope calculation for the circular analysis, we have also reversed the order of the figures and discussion of these issues, so that the regression slope now becomes Fig 2 and the circular plot becomes Fig 3.

The new circular plots (Fig 3) have now been replotted based on a PDM of 198°, and all associated results and statistics changed in the results section at lines 356-363.

The authors state that the prevailing winds are the same in autumn and spring, but examination of figure 1 suggests that the prevailing wind is from the southwest in spring and more westerly in autumn. Could they test their assumption that the winds are consistently from the southwest (l. 104) at the sites of the vertical looking radar?

***** Our Response:** We agree, and thank the referee for noting this. Patterns of downwind directions on all days were very similar between locations within the same season (see the new Tables S1 & S2 and Figures S2 & S3 in the supplement), but as indicated by the referee there was a significant (albeit small) difference between seasons – from SWS in spring (downwind direction = 60°, towards ENE) but more from W in autumn (downwind direction = 84°, towards E). We have made this change in the results section at lines 281-290, and added the plot of winds on all days to Figures 1, S2 and S3 to show the small seasonal difference in wind direction. This change does not affect the results in any

way, as the wind directions of non-mass migrations were the same (73 vs 76°) and the wind directions of mass migrations were clearly different as already discussed.

The authors take directions of multiple individuals and combine them into a mean track and heading direction for each migratory day. However the mean orientations have not only a direction to the vector but also a length that varies with concentration, and so when analyzing the mean of these means, second order circular statistics should be applied (see Zar or Batschelet for methods).

***** Our Response:** Thank you for this very helpful suggestion. In fact, Referee 2 also raised a similar point about how to analyse data from daily means, and so please see our response and associated changes to these analyses in the response to Referee 2.

This may also be true for the wind measurements, but it is unclear how many directions are collected each day.

***** Our Response:** This is not an issue for wind measurements, as there was an identical sample size (8 hourly values) on every day.

In figure 3, the values on the y-axis are compass degrees, which are not linear. Rather they are circular, and the authors need to apply circular statistics to these data rather than the linear regressions that they used. Alternatively, linear regression can be applied if the compass directions are limited to a semi-circle (180°, see Zar).

***** Our Response:** We agree, and have taken the advice to do the second method you suggest – we have restricted the compass directions analysed in this method (was Figure 3, now Figure 2 in the revision) to the linear section of the data (from 90 to 270° in autumn, -90 to +90° in spring). This made no difference to the spring result (which is still not significant), but has changed the autumn result, so the PDM and slope now have a different value (see new Fig 2 and lines 338-346 in results). As this also changed in response to the comment from referee 2 about taking account of sample size using a weighted regression, we discuss the implications of these changes to the PDM and slope in the response to Referee 2 (see below).

Specific comments:

I. 327 although the potential for modest partial compensation has not been eliminated by the CI (0.75, 1.12).

***** Our Response:** Due to the change of analysis of this data, our slope of 1.18 (95% CI 1.05 – 1.30) no longer is consistent with partial compensation or constant compass course, and so this comment is no longer applicable. We have changed the text at lines 338-346 in accordance with this change, and discuss the implications of this in response to Referee 2 below.

I. 344 There is no difference in the winds on non-migratory days but winds are significantly slower on autumn migratory days. Doesn't there have to be a different distribution of winds from which to select for this to be true? Is the distribution of wind speeds in spring and autumn the same?

***** Our Response:** There is a small difference in the mean wind speed (spring = 8.27 m/s, autumn = 8.19 m/s) as suggested, but in fact this difference was not significant (Mann-Whitney U-test, $P > 0.05$). We have added some text to the results at lines 372-373 to show this, and also included a new Figure S7 with the total wind speed distribution.

I. 377 If they were simply flying downwind, then axis of orientation would equal wind direction.

*** **Our Response:** Good point, we agree that this was not worded clearly enough in the original draft. We have changed the text at lines 409-414 to indicate our ideas more clearly, as follows:

“The strategy in the spring therefore appears to be one of aligning their heading vector relatively close to the downwind direction (albeit with a fair degree of error), but without any evidence for compensatory orientation in the direction of their heading offsets with respect to the PDM. When this is combined with the positive influence of warmer winds on the probability of initiating migration, this strategy should be sufficient for a general northward expansion during the spring [9].”

I. 382 ‘Restriction of behavior’ may be too extreme a term. They may actually attempt to migrate but find it to no avail and drop out. As a result, migrating insects are rare and observed less or excluded as too few to be deemed a mass migration in unfavorable winds.

*** **Our Response:** We agree, and have modified the text (please see lines 417-419).

I. 412 Although a compass might provide a general direction, it may not need to be time compensated. Leading lines are a real possibility for maintaining a common orientation throughout the daytime.

*** **Our Response:** We deal with the issue of time compensation in a response to a comment from referee 2, please see below.

Referee: 2

Major concerns

1) In the abstract (line 34) and discussion (lines 411 – 417), the authors assume hoverflies employ time compensation. However, I think it should be possible to test this idea more explicitly using the data at hand - an exercise that would improve the depth and impact of the paper. In particular, I should be possible to parse the autumn mass-migration events into a.m. and p.m. epochs (based on the time course of the raw VLR observations - not the daily means) and then regress the average tracks on the average alpha value within the two datasets. If these hoverflies employ time compensation, these two regression lines should have very similar intercepts. On the other hand, if hoverflies instead maintain fixed (uncompensated) headings with respect to the moving sun, one would expect shifted intercepts for a.m. versus p.m. data (see drawing at right).

*** **Our Response:** Thank you for this excellent suggestion. We have carried out the suggested analysis, and included the new data in the Results at lines 342-348 and in the new Figure S6 in the supplement. The results show that the PDM is very similar in the morning (197°) to the afternoon (201°), and to the overall dataset (198°), and none of these values are significantly different from each other. We therefore conclude that our results are consistent with the use a time-compensated sun compass, and include the following modified text in the discussion at lines 459-467:

“For seasonally-directed migrations like that exhibited by hoverflies, a time-compensated celestial compass is required for navigation in a beneficial direction (i.e. towards SSW in the autumn), and this has not yet been demonstrated in the Diptera. It is a well-known feature of butterfly migration [17,35] however, and thus we assume that a time-compensated sun compass (based on the position of the sun, polarized skylight, and/or the chromatic or

intensity gradient of the sky) will prove to be the mechanism used by migrant hoverflies. Our finding that the PDM remains constant between the morning and afternoon (see figure S6), is consistent with the use of a time-compensated sun compass.”

It may also be worth noting that the hoverflies might not appreciably benefit from the increased control over heading provided by time compensation. Figure 1 in Guilford and Taylor’s 2014 paper provides an illustration of theoretical headings adopted, over the course of a day, by a migrant employing celestial navigation with and without time compensation. Their figure suggests situations in which a migrant lacking a time-compensation mechanism may yet be able to maintain an acceptably well-regulated heading; for example, if the migrants only fly for a few hours on any given day, the angular deviation of their self-powered trajectory would be modest.

*** **Our Response:** Thanks, this is an interesting suggestion, but as we have now found evidence to support the existence of time-compensation, we don’t feel it is necessary to discuss this situation.

2) I have some concerns with the fact that, in all analyses, the radar data were first compressed into daily means (mean tracks, mean headings) across individual hoverflies. I understand that regressing track angles on α is a powerful way to assess migration strategies, given that this method capitalizes on the variability in wind conditions among mass-migration events, and does not require the destination [PDM] to be specified a priori (Green and Alerstam 2002). From reading Green and Alerstam’s work, I understand that a regression analysis performed on individual migrant data can yield spurious results if the dataset does not derive from a uniform circular distribution of wind directions. Specifically, their modeling indicates that datasets dominated by a broad range of tailwinds (a scenario obviously applying to the manuscript at hand) are prone to falsely report a behavioral strategy of overcompensation. I assume this explains why the authors have decided to report means for each mass migration event. If this is true, I think it would help future readers to say something more explicit about this justification. Until digging extensively into Green and Alerstam (2002), I was perplexed as to why the paper deals only with daily means.

That said, I still think that some possibly important information is lost by only considering daily means. For example, when first looking at Figure 3, I wanted to know which scatter points represented a mass migration of, say, many hundreds of individual hoverflies, and which represented far fewer observations. For Figure 3b specifically, I wondered how the regression slope might change if the fitting procedure was weighted to account for the number of raw observations underlying each point. Might outlier points represent daily migrations with comparatively few flies? By squinting at Figure 3a, one can see a possible underlying upward trend in the data, which might be significant if the outlier points did indeed represent situations based on relatively few individuals. In any event, I think the authors would be well served by addressing this issue.

*** **Our Response:** This is a very good point, and we have now re-analysed the track vs α plots (now Figure 2 in the revised paper) using the weighted least squares linear regression method, to take account of the variability in the daily sample sizes, as requested (please see lines 242-248 in the methods). In addition, in response to Referee 1, we also restricted these analyses to the linear (180°) subset of the data. With these new plots, the spring dataset remained non-significant, so there was no change there. However, in autumn, the combination of these two changes made a modest, but important, change to both the slope and the PDM.

Firstly, the PDM changed to 198° (with 95% CI of 193-202°), and this was no longer very close to 180, we decided to use the value of 198 for our autumn PDM when calculating the degree of correction for drift in the circular plots (now Fig 3). Note, because we now use a calculated value for the autumn PDM which originated from this linear regression, we have reversed the order of the original Figures 2 and 3, and the order in which they are discussed in the results. The change of the PDM to

198 (from 180) did not substantially affect the results of the circular analyses on the degree of correction, which still showed no correction for drift in spring, but a large (and highly significant) correction for drift when the drift angle was $>10^\circ$.

Secondly, the slope of the linear regression changed – the original value of 0.94 (95% CI from 0.75 to 1.12) was consistent with a strategy of Constant Compass Course, but the new value of 1.18 (95% CI from 1.05-1.30) is significantly greater than 1. According to Green & Alerstam, this would indicate their category of “overdrift” (which, in bird terms was seen as a kind of error), but we have previously argued (e.g. Chapman et al 2011, *Current Biology*; Chapman et al 2016, *J. Animal Ecology*) that in fact this represents an adaptive strategy for organisms which value speed over direction, but which still want to influence their direction somewhat; and it is also the same orientation strategy as used by Silver Y moths during the autumn (e.g. Chapman et al 2008, *Current Biology*; Chapman et al 2010, *Science*).

We have therefore extensively modified the text in both results (see lines 334-342 & 356-363) and discussion (see lines 425-451) to reflect these new results.

I also wonder why daily means were used Figures 1 and 2. Why not simply present circular histograms of all individual data, pooled over mass-migration events, for spring and autumn? The authors could consider using a ranked Rayleigh test (Moore 1980) to account for varying vector lengths of each mass-migration event, but to my understanding this does not explicitly deal with variable sample sizes in each daily mean. Would be more straightforward to run a single Rayleigh test on all individual data? As mentioned above, analyzing individual observations (and when they were made throughout the day) might provide a means of testing for time compensation.

***** Our Response:** We have followed this suggestion, and analysed the mean track and heading directions of all individual hoverflies in spring and autumn. The values from these new analyses were very similar to the values from the daily means, with all differences in pair-wise comparisons falling between 0 and 8° (spring track 342 vs 335; autumn track 180 vs 185; spring heading 314 vs 322; autumn heading 198 vs 198). We feel that this gives strong support to the idea that analysis of daily means as we used in the original draft is a suitable way to present the data, and that using individual data (or using the modified/ranked Rayleigh test as also suggested by Referee 1) will not make any substantive changes to the results. We have added this information to the results at lines 280-281, and included the plots as a new figure in the supplement (see Figure S5).

We also repeated the analyses for the circular plots of the degree of correction (was figure 2, now figure 3), and again the results were not changed in any substantive way (all mean correction values calculated from the individual data were within a few degrees of the same values from the daily means), but for simplicity's sake we have not included these extra values and figures in the paper.

3) I think the filtering scheme used to select for hoverflies (selecting a range of body masses and reflectivity shape value ratios) is insufficiently described in this paper, given its importance for all subsequent analyses. I do appreciate that this procedure is clearly grounded on prior work (Wotton 2019). However, given the important this supporting information ought to be concisely summarized in the methods section of this present manuscript as well.

***** Our Response:** We have substantially increased the amount of detail on our filtering procedure, please see lines 144-161 in the methods.

Furthermore, I think it would be helpful for the authors to report pre-filtered versus post-filtered insect counts, for both “mass-migration” and “non-mass-migration” events. Knowing what percentage of all individual observations were classified as hoverflies (and what percentage were not) would be very helpful in interpreting both the methodology and the underlying natural phenomena. This should be provided in the text.

***** Our Response:** The post-filtered counts of hoverflies are presented in Table S1. We have now added the percentage of all insects which were classified as hoverflies in the methods, please see lines 173-177.

A related point concerns the question of whether a hoverfly’s body pitch influences the $\sigma_{xx} / \sigma_{yy}$ ratio generated by its body (and I certainly defer to the authors’ expertise with radar data). Could this ratio vary as the cosine of the insects’ body pitch relative to the horizontal? I was curious about this point because there is ample evidence that flies alter body pitch as one mechanism to modulate forward airspeed – which might be relevant to data collected under different wind conditions. Is there any chance that observations might be mis-classified (either false negatives or positives) depending on the body pitch of the insect?

***** Our Response:** This is a very interesting idea! We had not considered it before, but fortunately, this spring, a member of our research group (Will Hawkes) collected numerous videos of actively-migrating hoverflies in Cyprus (during a mass invasion of the hoverfly *Eristalis tenax*). Will extracted screenshots of 100 different individuals and measured the body angle of the migrating hoverflies relative to the ground, and found that the hoverflies, without exception, were migrating with a completely horizontal body (mean difference between body pitch angle and a perfectly horizontal reference line was 0.3° , with a circular standard deviation of 1.42° and r-value of 0.996). We have added this information to the methods (see lines 161-167), and some screen grabs of migrating hoverflies as supplemental figure S1, and added Will Hawkes to the authorship to recognise his input.

4) It would be helpful to present, in or alongside Figure 3b, mean headings also regressed on mean alpha values. My understanding of Green and Alerstam’s regression method for assessing drift strategies (2002) is that a full drift (or constant heading) strategy is characterized by a track/alpha slope of one, and a heading/alpha slope of zero. This manuscript presents evidence in support of the former, but seemingly not of the latter.

***** Our Response:** Because the new way of analysing the data in the regression plots (now Figure 2) led to a change in the slope (and thus the orientation strategy) in the autumn dataset, we no longer have a situation where we expect to see a slope of zero if we carry out the same analysis for headings. We have plotted this data and we get a slope of 0.30 (95% CI of 0.19 and 0.40), but due to the complexity of the analyses already presented, we are not convinced that adding this extra plot to the paper will help the reader follow the arguments. We have therefore decided not to include this extra plot, but if the referee or editor feels strongly that we should include it, we can do so at the next stage of revision.

Minor comments

Figure 2: The sub-analyses in Figure 2 (b) and (d) are mostly lucid, but one aspect of these circular histograms is confusing: the overall skylines of the histograms (panel a versus panel b, and panel c versus panel d) appear to have been altered by the process of fractioning the datapoints into those with track $> 20^\circ$ from PDM and track $< 20^\circ$ from PDM. I am guessing that the histogram binning was

somehow perturbed by this layer of analysis and think addressing this would improve the figure's clarity.

***** Our Response:** Actually this is not an effect of the binning procedure. It is because when the total dataset is plotted in the left-hand columns, all data points in the same bin are 'stacked' on top of each other, but when they are fractionated into the two categories of data in the right-hand column, those data points from one category ($<10^\circ$) are not stacked upon those from the other category ($>10^\circ$) when they happen to fall in the same bin, but are placed in front of them. This results in a changed 'skyline' when visually comparing the plots, but it is not due to the binning procedure. Note, this is now Figure 3 in the revision, and the threshold for the two categories changed from $<20^\circ$ to $<10^\circ$ because the change of PDM from 180° to 198° led to a change in the pattern of correction.

Figure 3: The analysis in Figure 2 was conducted with respect to an assumed autumn PDM of 180° , but Figure 3's regression analysis revealed the PDM to be 189° . Does the finding from Figure 2 change appreciably if the updated estimate of PDM is used?

***** Our Response:** See response to Referee 1 – the PDM actually changed to 198 using the new method of calculating it, and we have used this value instead of a value of 180.

L36: "...arriving in the UK in spring, showed less orientation ability..." I suggest phrasing this more phenomenologically, because its current wording raises some questions about orientation mechanisms that the paper does not go on to address. Perhaps "... showed weaker orientation tendencies" or even "... showed a lesser degree of orientation".

***** Our Response:** we have changed the text to read: "showed weaker orientation tendencies", see line 36.

L316: "This result indicates that as the degree of ... drift away from the seasonal PDM ... increased, that hoverflies showed a greater offset..."
Omit one of the two "that's" in this sentence.

***** Our Response:** Done.

L320: "... assigned a positive or negative sign depending on whether the heading was to the left or right of the track..." I found the left/right designation confusing for circularly distributed data.

***** Our Response:** we have changed the "left / right" designation to "anti-clockwise" and "clockwise" (see lines 232-233), and also refer to figure S4 to help readers understand our methods.

L323: "... spring hoverflies make no attempt to correct for drift." and L328: "[in autumn hoverflies keep a] 'constant compass course' irrespective of the wind "

The distinction between these two inferred behavioral strategies ought to be made clearer by editing line 323. At first glance "making no attempt to correct for drift" and "full drift" sound very similar, but the latter refers to a distinct strategy in which mean heading is fixed and wind drift fully interacts with the fixed heading to produce the track.

***** Our Response:** Because the autumn hoverflies no longer show a strategy of constant compass course (full drift), this potential for confusion has been removed, and so we have made no edits to the text here.

L339: "...95% of occasions with speeds between 4 and 9 m s⁻¹ (figure 4a)..."

The description of the box/whisker plots in Figure legend 4a suggests that actually only 50% (the interquartile range) of mass-migration occasions had wind speeds between 4 and 9 m s⁻¹.

*** **Our Response:** Thanks for noting this mistake, we have made the change.

L547 (figure legends): "... mass migration or a non-migration event."

I think the authors meant to write "non-mass-migration event."

*** **Our Response:** Thanks for noting this mistake, we have made the change.

We hope that the changes we have made will satisfy the referees and editors, and we look forward to hearing back from you.

With kind regards,

Boya Gao and Jason Chapman, on behalf of all co-authors.

Appendix B

To: the editorial team of *Proc Roy Soc B*

From: Miss Boya Gao & Dr Jason Chapman, University of Exeter

30 April 2020

Ref: revision of ms RSPB-2020-0406

(Gao et al: Adaptive strategies for high-flying migratory hoverflies)

We wish to thank the editorial board and the referee for providing further helpful comments on our revised submission, and allowing us another opportunity to resubmit. We provide detailed responses to each of the referee's comments from the second round of revision below. The new text in the manuscript resulting from this second round of revision is highlighted in red font to aid assessment of our changes. We hope the editorial board and referee will agree that our revised version of the paper is much improved; this is largely due to the helpful and constructive suggestions from both referees during the two rounds of review, and so we thank them for their valuable input.

With kind regards,

Boya Gao and Jason Chapman, on behalf of all co-authors.

Responses to comments from Referee 1:

I have reviewed this paper before and I appreciate the authors attention to improving the manuscript. My principal suggestion is a more detailed analysis of time compensation, why I detail in the minor comments below.

*** **Our Response:** We thank the referee for the very helpful suggestions surrounding the time compensation analyses, and we have now done the new analyses recommended by the referee and included them in the revised paper – please see the details below.

There is still a lot of repetition between the Introduction and Discussion which would shorten the Discussion if revised.

*** **Our Response:** we have carefully edited the discussion to shorten it and tighten the focus.

In addition, it remains unclear what data has been published in the previous paper on the topic. Is it all of Figure 1, or Figure 1a?

*** **Our Response:** It was only Fig 1a (the two plots showing migration track direction), the rest of Fig 1 is completely new. We feel it is essential to include the track plots, otherwise it is impossible to understand the rest of the analyses, but we have moved the text describing these results from the Results section to the Introduction (see lines

104-107) to make it clearer that this part of the data is repeated from elsewhere, and we also explicitly state this in the figure legend for Figure 1a. In addition, the results of track directions based on all individual hoverflies (not daily means), which are included in Fig S5, are new and not reproduced from the earlier study, so we retain mention of these values in the Results section at lines 284-287.

I. 101 change 'migrate' to 'fly' since migrate repeats migrant

*** **Our Response:** we have made the change as suggested.

I. 108 Lines 276-281 on directions of migrants should be moved to this paragraph which provides background information

*** **Our Response:** Ok, we agree, we have moved a summary of the directional information to the introduction to make things clearer, please see lines 104-107.

I. 252 change 'they were' to 'it was'

*** **Our Response:** done.

I. 266 state how many records per day

*** **Our Response:** Good point, it has been added at line 273.

I. 269 change 'was' to 'were'

*** **Our Response:** done.

I. 281 It remains completely unclear what is new and what is previously published. By moving published material to the Introduction.

*** **Our Response:** Apologies that this has remained unclear. The only material that is previously published is the distribution of track directions based on daily means, which is shown in Fig 1a. The rest of Figure 1 (and incidentally the track direction values based on individual flies, shown in Figure S5) is all completely new and previously unpublished. We have made this totally clear now by modifying the text in the Introduction at lines 104-107, the results at lines 284-287, and in the legend of Figure 1a (all revised text highlighted in red for ease of checking).

I. 284 References to an 8-point compass are very confusing. 84 degrees is actually closer to ENE than 61 degrees. Why not just state the degrees? Or refer to the first was as 'more northeasterly' and the second one as 'more easterly'.

*** **Our Response:** We accept that reference to a 16-point compass can be confusing, and so we have modified the text here by simply mentioning the actual direction.

I. 290 In the text you compare the orientations to the total data set, but the statistic compares the two orientations. The text and the statistical test should agree. Which comparison is more important?

*** **Our Response:** Thanks for pointing this out; on reflection, we agree that we should not have used the Watson-Wheeler test to compare wind directions on

non-migration days with the total dataset, as there is a large overlap in the data-points making up these two distributions; we therefore removed mention of this test here, and just compare the mean directions; please see lines 292-297 for revised text. We now restrict use of the Watson-Wheeler test to compare the wind distributions on days with migration versus days without migration, as there is no overlap between these datasets (line 300 in spring, and line 306 in autumn).

I. 292 I thought it was ENE

*** **Our Response:** We no longer mention ENE as we have removed usage of 16-point compass directions as requested.

I. 337 Move the 95% C.I. to just after slope of 1.18 and delete 'do not overlap'

*** **Our Response:** We have made this change (line 342).

I. 338 If 1 is full compensation and 0 is no compensation, then greater than 1 is 'overcompensation' for wind drift, which means the insects are turning further into the wind than necessary to compensate for wind drift. It's unclear how you can call this 'compass-biased downstream orientation.'

*** **Our Response:** Actually, that is not the case; a slope >1 indicates the strategy which is called "overdrift" by Green & Alerstam (2002); 'overcompensation' is indicated by a regression slope <0 . When Green & Alerstam coined this phrase in their 2002 theoretical paper, the "overdrift" strategy had never been observed in the birds they were studying, and they included this possible outcome in their framework for the sake of completeness, but gave it a rather unhelpful meaningless name which indicates it is an orientation error rather than a beneficial strategy. This was because they were very bird-orientated, and thinking about directional control rather than ground speed. When we discovered this strategy in moths in our 2008 and 2010 papers in *Current Biology* and *Science*, we showed it is actually highly beneficial for short-lived insects where speed (with some directional influence) is more important than tight directional control. We therefore teamed up with Thomas Alerstam to write a new framework of orientation responses to currents across all taxa (not just birds), where we renamed the "overdrift" strategy as "*compass-biased downstream orientation*", which we published in Chapman et al 2011 *Current Biology*. In this strategy an organism deviates its heading slightly from the downstream direction so that it lies between the downstream and the seasonally-favourable direction, thus maximizing ground speed (and distance covered) while still influencing direction in a beneficial manner. We have used the term in many subsequent papers studying nocturnal insect orientation, and now, for the first time, we show that this strategy is also used by day-flying insect migrants. We describe this strategy in the methods at lines 244-245 and in the discussion at lines 442-451.

I. 342 There are several ways to compensate for wind drift. Separation of the data into morning and afternoon can not distinguish between these mechanisms. For example, compensation in the morning and afternoon can be achieved with a step

function time compensation mechanism such as that possessed by naïve honeybees. Time averaging and full compensation can be investigated with hourly averages. See Oliveira et al. 1998 JeB or the chapter by Srygley and Oliveira in *Insect Movement* (2001, Fig. 9.3) for the method.

*** **Our Response:** We appreciate this excellent suggestion from the referee, and have done the new analyses as requested. We calculated hourly values of both track direction and heading throughout the day, and compare them with the overall daily values for each measure (180° and 198° respectively) in the new Figure S6 in the supplement (with all data values contained in the new Table S4). They show very little departure from the daily values (the dashed horizontal line), and no significant trend with time in both cases. We therefore conclude that this shows evidence for continuous compensation for the sun's motion. We include this new information at lines 348-356, and add the Oliveira et al (1998) reference.

I. 346 Shouldn't 'autumn' be 'afternoon' in this sentence?

*** **Our Response:** Thanks – yes, it should have been 'afternoon' instead of 'autumn', but this text has been removed in any case due to our new hourly analyses.

I. 384 The Discussion stills seems repetitive of the Introduction.

*** **Our Response:** we have edited the discussion to remove repetitive text, and have now decreased the word count in this section from 1455 to 1274 words.

I. 428 The following statement is not true: Insect migrating close to the ground have complete control over their flight direction, and thus their ground direction can be assumed to be the same as their PDM. There are instances when drift would be the optimal strategy, and in those instances ground direction is not the same as the PDM.

*** **Our Response:** We agree with that in some situations this statement is not true, so we changed the wording as follows: "Insects migrating close to the ground have greater control over their flight direction than high-flying migrants, and thus we assume their ground direction is likely to be similar to their PDM."

I. 438 'Employment'

*** **Our Response:** Thank you, we have made the change.

I. 460 a time-compensated celestial compass is not required. Only a global orientation cue. So magnetic compass would suffice. In this sentence, you really just want to focus on the time-compensation part as the 'celestial compass' has already been discussed in I. 456.

*** **Our Response:** Good point. We have revised this text at lines 460-466 to make this clearer.

I. 466 Change 'an' to 'and' You've only shown limited time-compensation. You should do the complete analysis that I suggest.

*** **Our Response:** Thanks - our new analyses, as suggested by the referee, do

indeed provide better evidence of full compensation for the apparent motion of the sun.